# Conformational biosensors delineate endosomal G protein regulation by GPCRs

Brian Wysolmerski [1,2], Nicole M. Fisher[1], Andrew N. Dates[1], Asuka Inoue[3,4], Emily E. Blythe [1,5] ✉ & Mark von Zastrow [1,6] ✉

Many GPCRs trigger a second phase of G protein-coupled signaling from endosomes after signaling from the plasma membrane, necessitating GPCRs to increase the concentration of active-state G proteins on the endosome membrane. How this is achieved remains unclear. Here, we show that three $G_s$-coupled GPCRs–the β2-adrenergic receptor, VIP-1 receptor, and adenosine 2B receptor–each trigger a net redistribution of $G\alpha_s$ from the plasma membrane to endosomes at native expression levels and without requiring receptor internalization. We then show that active-state $G\alpha_s$ production on endosomes, in contrast, is GPCR internalization-dependent. We further identify location bias in the selectivity of GPCR coupling between $G_s$ and $G_q$ on endosomes relative to the plasma membrane. We propose that endosomal $G_s$ regulation involves discrete GPCR-G protein coupling reactions, one at the plasma membrane controlling $G_s$ concentration and another at endosomes controlling $G_s$ activity, and that GPCR endocytosis can switch signaling selectivity between G protein classes.

G protein-coupled receptors (GPCRs) constitute the largest family of signaling receptors and regulate nearly every physiological process. After activation by binding an agonist, GPCRs initiate signaling by coupling to cognate heterotrimeric G proteins, consisting of a Gα subunit and Gβγ subcomplex, which function collectively as key transducers of downstream signaling. This allosteric coupling reaction promotes guanine nucleotide exchange on the G protein α-subunit, resulting in GTP binding to the α-subunit that converts it from an inactive to active state. G protein classes are defined according to the identity of their α-subunit (e.g., $G_{s/olf}$, $G_{i/o}$, $G_{q/11}$, and $G_{12/13}$), with individual GPCRs differing in selectivity for coupling among G protein classes that produce distinct downstream regulatory effects[1,2]. The central importance of GPCR signaling in physiology, as well as its dysfunction or dysregulation in a variety of pathological states, has motivated intense interest in GPCRs as therapeutic targets, and presently over 30 % of FDA-approved drugs target GPCRs[3].

The importance of GPCR signaling from the plasma membrane has been recognized for many years[1,4], and there is now considerable interest in the ability of GPCRs to produce distinct and additional effects from intracellular membranes[4–6]. Endomembrane signaling is perhaps most strongly supported from the study of $G_s$-coupled GPCRs[4,5,7–17], but there is also significant evidence for endomembrane signaling through other G protein classes as well[5,18,19]. Based on first principles, such signaling fundamentally depends on the presence of active-state G proteins on the appropriate membrane[1,2]. However, it remains unclear how the concentration of active-state G proteins on endomembranes is regulated.

Under basal conditions, $G_s$ is enriched on the plasma membrane and present in lower amounts on intracellular membranes[20]. $G_s$ activation by coupling to a GPCR on the plasma membrane promotes dissociation of $G\alpha_s$ and its net intracellular redistribution, increasing the concentration of $G\alpha_s$ on multiple endomembrane compartments,

[1]Department of Psychiatry and Behavioral Sciences, University of California, San Francisco, San Francisco, CA, USA. [2]Tetrad Graduate Program, University of California, San Francisco, San Francisco, CA, USA. [3]Graduate School of Pharmaceutical Sciences, Tohoku University, Aoba-ku, Sendai, Miyagi, Japan. [4]Graduate School of Pharmaceutical Sciences, Kyoto University, Yoshida-Shimo-Adachi-cho, Sakyo-ku, Kyoto, Japan. [5]Department of Genetics, Cell Biology, and Development, University of Minnesota, Minneapolis, MN, USA. [6]Department of Cellular and Molecular Pharmacology, University of California, San Francisco, San Francisco, CA, USA. ✉e-mail: eblythe@umn.edu; mark.vonzastrow@ucsf.edu

including endosomes[20–24]. The production of active-state $G\alpha_s$ on endosomes is then thought to require a second coupling reaction occurring locally on the endosome limiting membrane[4,5,10]. There is considerable evidence supporting such activation (e.g[8–11,16]), but the presence of active-state, GTP-bound $G\alpha_s$ on endosomes has not been directly demonstrated. Accordingly, the subcellular location(s) of G protein activation by GPCRs, and of active-state $G\alpha$ subunit accumulation on endomembranes, remain incompletely understood.

Here, we addressed this knowledge gap by dissecting the regulation of endosomal $G\alpha_s$ localization and activity by GPCRs. We first verify that the prototypical β2-adrenergic receptor (β2AR) triggers a rapid intracellular redistribution of $G\alpha_s$ from the plasma membrane[20,21,23–25]. We then extend the present understanding by showing that this process is triggered by a variety of $G_s$-coupled GPCRs, and under physiologically relevant conditions of native or near-native levels of GPCR and G protein expression. Next, using conformational biosensors, we demonstrate sequential phases of both $G_s$ activation and active-state $G\alpha_s$ accumulation, first on the plasma membrane and then on endosomes, and at endogenous levels of G protein expression. We then show that the accumulation of active-state $G\alpha_s$ on endosomes is specifically dependent on receptor endocytosis. Finally, we provide evidence for a type of location bias in the biochemical selectivity of endosomal G protein activation that is programmed by GPCRs in a receptor-specific manner.

## Results

### $G\alpha_s$ colocalizes with internalized receptors and Gβγ on early endosomes after GPCR activation

$G\alpha_s$ associates with various cellular membranes and is enriched on the plasma membrane in unstimulated cells[20–25]. It is generally thought that $G\alpha_s$ dissociates from the plasma membrane after its activation there and subsequently samples a variety of intracellular membrane compartments, including endosomes[20,23,24,26,27], where a second round of GPCR-triggered activation occurs[4,5,8,10,11]. We sought to verify and further examine this fundamental signaling process in living cells. We began by imaging $G\alpha_s$ by confocal fluorescence microscopy, using β2AR as a model $G_s$-coupled GPCR that is well known to trigger intracellular redistribution of $G\alpha_s$ and has traditionally been used to study it[20,21,23–25]. We labeled $G\alpha_s$ by inserting EGFP into the linker region between its α-helical domain and conserved Ras-like domain, a strategy shown previously to preserve the signaling function of $G\alpha_s$[25], and then expressed this labeled construct in HEK293 cells stably expressing Flag-tagged β2AR. Consistent with previous studies[21,23–25], EGFP-$G\alpha_s$ visibly redistributed intracellularly from the plasma membrane within several minutes after application of isoproterenol, a β2AR agonist (Iso, Fig. 1a). EGFP-$G\alpha_s$ was diffusely distributed in the cytoplasm and concentrated on various endomembranes, including endosomes, as indicated by colocalization with mApple-EEA1 (Supplementary Fig. 1a). We previously described Iso-induced intracellular redistribution of an epitope-tagged $G\alpha_s$, but not specific localization to endomembranes, in formaldehyde-fixed cells[21]. Potentially explaining this, we found such fixation to incompletely preserve endomembrane association of EGFP-$G\alpha_s$ (Supplementary Fig. 1b).

According to the present understanding, $G_s$ activation on endosomes would require receptors to be present in the same endosome membrane[2,4–6]. We tested this in our system by imaging EGFP-$G\alpha_s$ and internalized β2ARs. Confocal microscopy resolved EGFP-$G\alpha_s$ localization on Flag-β2AR-containing endosomes in isoproterenol-treated cells, and we also observed EGFP-$G\alpha_s$ localization on additional compartments not containing internalized receptors (Fig. 1a, arrows indicate examples of EGFP-$G\alpha_s$ / Flag-β2AR colocalization). We quantified colocalization using Pearson correlation analysis, assessing pixel-based correlations between EGFP-$G\alpha_s$ and either the Flag-β2AR or mApple-EEA1 fluorescence signal, respectively, after agonist application. The Pearson correlation between EGFP-$G\alpha_s$ and Flag-β2AR

decreased after agonist application (Fig. 1b), reflecting a loss of Flag-β2AR and EGFP-$G\alpha_s$ colocalization at the plasma membrane, while the correlation between EGFP-$G\alpha_s$ and mApple-EEA1 increased (Fig. 1c), reflecting $G\alpha_s$ redistribution to early endosomes. However, the correlation between EGFP-$G\alpha_s$ and Flag-β2AR plateaued at a level that remained significantly higher than that between a cytosolic EGFP control and Flag-β2AR (Fig. 1b), consistent with our observations of partial colocalization with receptors on endosomes. Together, these results indicate that activated β2ARs trigger EGFP-$G\alpha_s$ to rapidly redistribute from the plasma membrane to endomembranes, including to endosomes that also contain internalized β2ARs.

If $G\alpha_s$ redistributes to endomembranes by partitioning, we anticipated that inhibiting $G\alpha_s$ dissociation from membranes would reduce its accumulation on endosomes. We tested this prediction using a previously described $G\alpha_s$ / $G\alpha_i$ chimera that retains the ability to functionally couple to β2AR but is more stably attached to membranes by the addition of an irreversible myristoylation[20,22] (membrane-pinned $G\alpha_{s/i}$). As expected, the membrane-pinned $G\alpha_{s/i}$ construct did not detectably redistribute or associate with endomembranes in response to Iso (Fig. 1c, Supplementary Fig. 1a).

Efficient GPCR-G protein coupling requires $G\alpha_s$ to be associated with Gβγ[2]; thus we asked if Gβγ is present on the same endosomes using an mApple-labeled γ-subunit (mApple-G$\gamma_2$) coexpressed with Gβ1. Labeled Gβγ localized to the plasma membrane and multiple internal membranes, consistent with previous results[28–30]. This included membranes associated with EGFP-$G\alpha_s$ and containing internalized β2ARs (Fig. 1d, Supplementary Fig. 1c). Whereas endosomal localization of both $G\alpha_s$ and β2AR were clearly increased after β2AR activation by Iso, the localization of labeled Gβγ was not noticeably changed (Fig. 1d). These results support a model in which $G\alpha_s$ and β2AR colocalize on endosomes in an activation-induced manner, while these endosomes appear to be constitutively associated with Gβγ[28]. Accordingly, all of the protein components necessary for G protein coupling converge at endosomes after agonist-induced activation of the receptor.

### $G\alpha_s$ returns to the plasma membrane after receptor inactivation

The ability of cells to respond to a subsequent agonist exposure would presumably require replenishment of $G\alpha_s$ at the plasma membrane, and previous studies indicate that the intracellular redistribution of $G\alpha_s$ triggered by β2AR activation is reversible after receptor inactivation[20,21]. We verified this in our hands by activating Flag-β2AR with Iso in cells coexpressing EGFP-$G\alpha_s$ and then applying the β2AR antagonist Alprenolol (Alp) in excess. We observed a pronounced reaccumulation of EGFP-$G\alpha_s$ on the plasma membrane after β2AR inactivation (Fig. 2a, Supplementary Fig. 2a), confirming reversibility of the $G\alpha_s$ redistribution process. We used nanoluciferase protein complementation (NanoBit) to quantify the reversible redistribution of $G\alpha_s$ by inserting LgBit into $G\alpha_s$ at the same position as EGFP (LgBit-$G\alpha_s$) and measuring complementation with a plasma membrane-targeted SmBit construct (SmBit-mApple-CAAX) verified to appropriately localize to the plasma membrane (Supplementary Fig. 2b). In this assay, intracellular redistribution of LgBit-$G\alpha_s$ is indicated by a decrease in the luminescence signal (Fig. 2b). Iso-induced activation of Flag-β2AR produced such a decrease with a similar time course as the redistribution observed by microscopy (Fig. 2c) and this recovered to baseline after adding Alp (Fig. 2c). These results indicate that GPCR-triggered intracellular redistribution of $G\alpha_s$ is indeed reversible, replenishing $G\alpha_s$ at the plasma membrane after activation is terminated.

### Intracellular redistribution of $G\alpha_s$ does not depend on GPCR endocytosis

Previous reports differ in whether the intracellular redistribution of $G\alpha_s$ triggered by β2ARs requires receptor endocytosis[21,23,24], so we

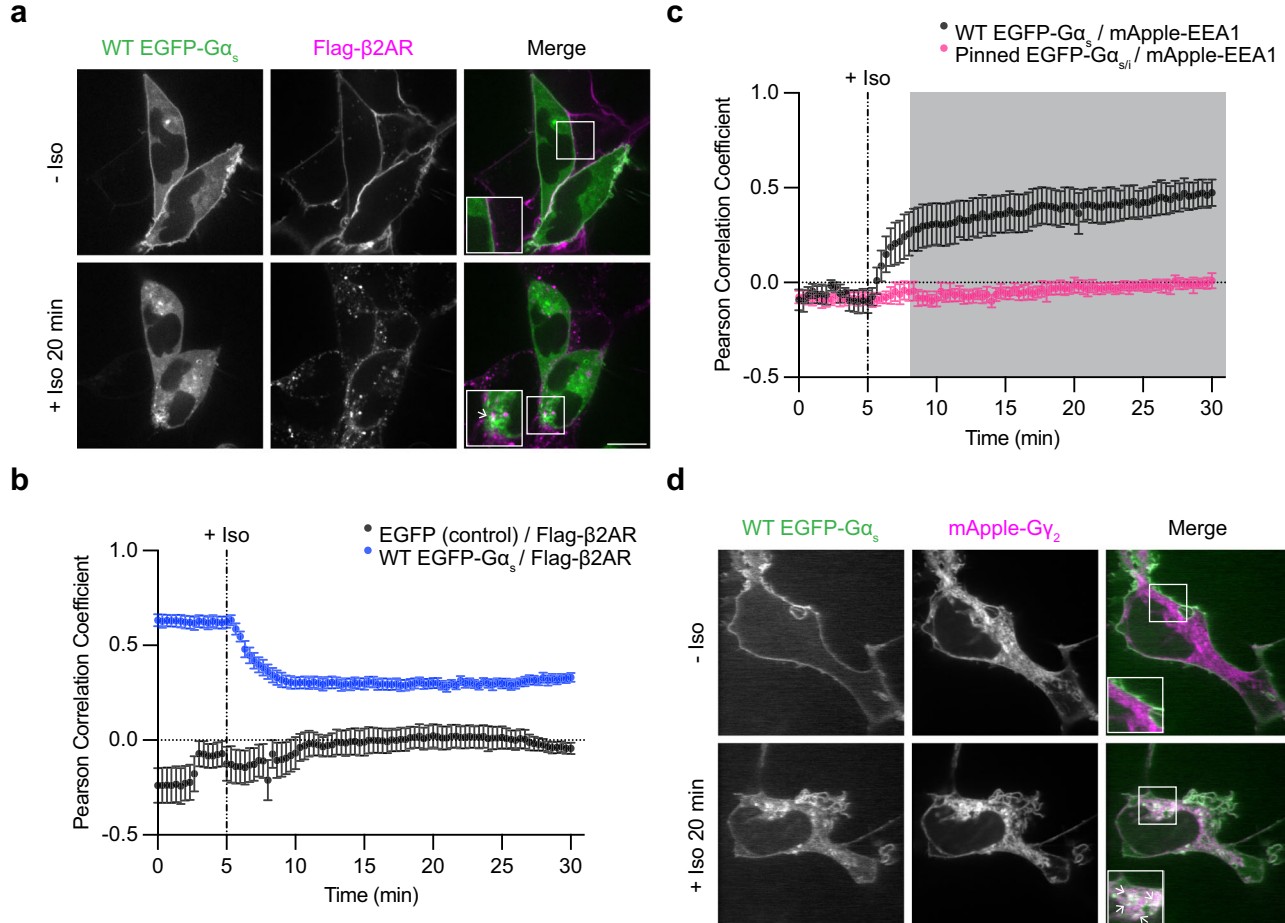

**Fig. 1 | Agonist-induced redistribution of Gα$_s$. a** Representative confocal images of live HEK293 cells stably expressing Flag-β2AR and transiently expressing EGFP-Gα$_s$ before and after 20 min of Iso (1 μM) treatment. **b** Pearson correlation coefficient between either the WT EGFP-Gα$_s$ or EGFP control channel and the Flag-β2AR channel over time. Significance determined by repeated measures 2-way ANOVA with Sidak's multiple comparisons test (see source data for *p* values). **c** Pearson correlation coefficient between WT EGFP-Gα$_s$ or membrane-pinned EGFP-Gα$_{s/i}$ chimera and mApple-EEA1 channels over time. Images are shown in Supplementary Fig. 1a. Shaded areas represent timepoints at which the difference between WT EGFP-Gα$_s$ and membrane-pinned EGFP-Gα$_{s/i}$ is statistically significant (*p* < 0.05). Significance determined by fitting a mixed-effects model using restricted maximum likelihood (REML) followed by Sidak's multiple comparisons test (see source data). **d** Representative confocal images of live HEK293 cells stably expressing Flag-

β2AR and transiently expressing EGFP-Gα$_s$ and mApple-Gγ$_2$ before and after 20 min of Iso (1 μM) treatment. Images are representative of at least three independent experiments. Prior to imaging in panels (**a**) and (**d**), cells were treated for 10 min with an anti-Flag antibody coupled to Alexa Fluor 647 to label surface Flag-β2AR. In panel (**a**), (**b**), and (**c**), cells were co-transfected with either myc-Gβ$_1$ and untagged Gγ$_2$, or untagged Gβ$_1$ and myc-Gγ$_2$. In (**d**) cells were cotransfected with untagged Gβ$_1$. Scale bars = 10 μm. Insets in panels (**a**) and (**d**) are 1.5x zoom of indicated regions and arrows indicate examples of colocalization. For Pearson correlation analysis in (**b**) and (**c**), data are represented as mean ± S.E.M. of individual dishes from at least three independent experiments (in **b**, *n* = 6 (EGFP control) or 16 (WT EGFP-Gα$_s$) movies and in (**c**), *n* = 7 (WT EGFP-Gα$_s$) or 8 movies (membrane-pinned EGFP-Gα$_{s/i}$)). Iso (1 μM) was added after 5 minutes of imaging, depicted by dashed lines. Source data are provided as a Source Data file.

further investigated this question using improved experimental tools. As a first approach, we acutely blocked β2AR endocytosis with the dynamin inhibitor Dyngo4a[31]. Iso-induced internalization of Flag-β2AR was strongly suppressed by Dyngo4a treatment, but the intracellular redistribution of EGFP-Gα$_s$ was not detectably affected (Supplementary Fig. 3a). As a second approach, we suppressed receptor internalization by mutating essential serine phosphosites in the cytoplasmic tail of β2AR to alanine[10,32,33] (β2AR-3S). HEK293 cells endogenously express β-adrenergic receptors[34], but at a sufficiently low level not to produce detectable Iso-induced Gα$_s$ redistribution signal in our NanoBit assay (Supplementary Fig. 3b). This enabled us to compare the effects of β2AR-3S to WT β2AR by simple overexpression after confirming comparable levels of expression (Supplementary Fig. 3c). β2AR-3S triggered a rapid and pronounced redistribution of Gα$_s$ from the plasma membrane indistinguishable from that triggered by WT β2AR (Fig. 2c), and this redistribution was accompanied by a corresponding increase in EGFP-Gα$_s$ accumulation on endomembranes (Supplementary Fig. 3d). As a third approach, we blocked β2AR

internalization using previously described β-arrestin 1/2 double knockout (*βarr1/2* DKO) HEK293 cells[16]. After verifying inhibition of WT β2AR internalization (Supplementary Fig. 3e), we found that Iso application triggered robust and reversible intracellular redistribution of Gα$_s$ in these cells (Fig. 2d, Supplementary Fig. 3e). Moreover, re-expression of βARR2-mApple in this genetic background fully rescued internalization of Flag-β2AR without detectably changing the redistribution of Gα$_s$ (Fig. 2d, Supplementary Fig. 3e). These results provide several lines of evidence indicating that GPCR-triggered intracellular redistribution of Gα$_s$ does not require internalization of the triggering GPCR.

## Intracellular redistribution of Gα$_s$ is triggered by multiple G$_s$-coupled GPCRs at native levels

We next asked if the ability to trigger intracellular redistribution of Gα$_s$ is shared by other G$_s$-coupled GPCRs. We focused on the adenosine-2B receptor (A$_{2B}$R or ADORA2B) and vasoactive intestinal peptide-1 receptor (VIPR1 or VPAC1) as representatives of GPCR family A and

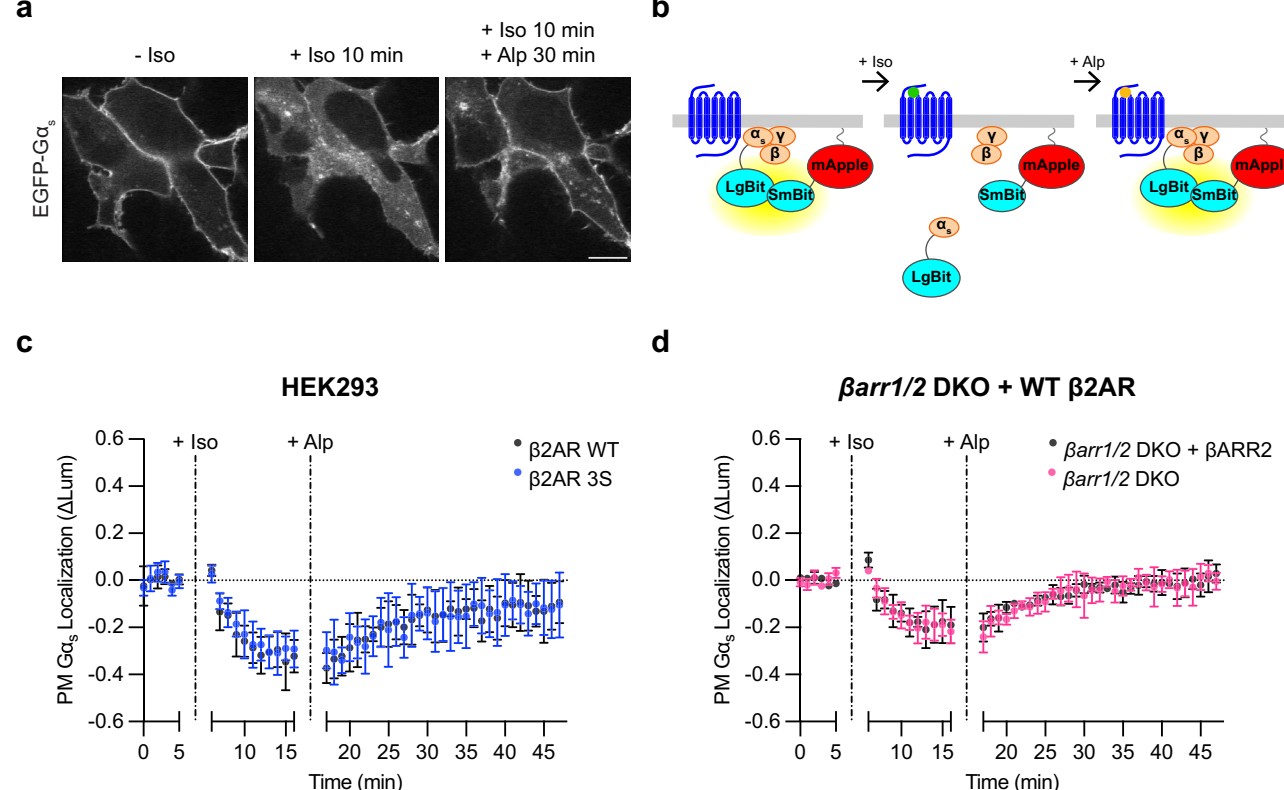

**Fig. 2 | Redistribution of Gα_s is reversible after receptor inactivation and independent of receptor endocytosis. a** Representative stills from time-lapse confocal microscopy of live HEK293 cells stably expressing Flag-β2AR and transfected with EGFP-Gα_s, myc-Gβ_1, and untagged Gγ_2 either before drug treatment, after 10 min of Iso (100 nM) treatment, or after 10 min of Iso followed by 30 min of Alprenolol (Alp, 10 µM) treatment. Images are representative of four independent experiments. Scale bar = 10 µm. **b** Schematic of plasma membrane Gα_s NanoBit bystander assay. **c** NanoBit bystander assay showing plasma membrane localization of Gα_s after Iso (100 nM at 5 min) treatment followed by Alp (10 µM at 16 minutes)

treatment in HEK293 cells expressing either Flag-β2AR WT or Flag-β2AR-3S. Significance (n.s.) determined by two-way ANOVA with Sidak's multiple comparisons test (see source data). **d** NanoBit bystander assay showing plasma membrane localization of Gα_s after Iso (100 nM at 5 min) followed by Alp (10 µM at 16 minutes) treatment in *βarr1/2* DKO HEK293 cells expressing either βARR2-mApple or mApple. Significance (n.s.) determined by repeated measures two-way ANOVA with Sidak's multiple comparisons test (see source data). Data are shown as mean ± S.D. of three biological replicates. Source data are provided as a Source Data file.

B, respectively, that are natively expressed in HEK293 cells and differ in their endocytic trafficking properties[34]. Specifically, VIPR1 robustly internalizes after activation through an arrestin-independent mechanism[16], β2AR undergoes arrestin-dependent internalization[35,36] and the human A_{2B}R is relatively resistant to agonist-induced internalization[34] (Supplementary Fig. 4). We found both VIPR1 and A_{2B}R to trigger a pronounced intracellular redistribution of EGFP-Gα_s when overexpressed and activated by cognate agonist (VIP or NECA), similar to the redistribution triggered by overexpressed β2AR after activation by Iso (Supplementary Fig. 4). These results indicate that multiple GPCRs can trigger redistribution of EGFP-Gα_s and provide further support for the conclusion that this process is not dependent on the trafficking properties of the triggering GPCR.

Because achieving GPCR-triggered redistribution of EGFP-Gα_s in these assays required the receptor to be overexpressed, we wondered if Gα_s redistribution requires supra-physiological expression of receptors or can be triggered also by native GPCRs. We reasoned that overexpressing EGFP-Gα_s in cells that already express endogenous, unlabeled Gα_s might limit the sensitivity of our assay for detecting redistribution triggered by endogenous receptors, particularly as Gα_s is endogenously expressed already at a much higher level than the GPCRs tested[29]. To address this, we expressed EGFP-Gα_s at a near-native level in cells lacking endogenous Gα_s. We first knocked out endogenous Gα_s by CRISPR-Cas9 editing of the *GNAS* gene (Gα_s KO cells) and verified full knockout genetically and by immunoblot, as well

as functionally by elimination of the Iso-induced cAMP elevation (Supplementary Fig. 5). We then isolated cell populations stably expressing EGFP-Gα_s at a near-native level, as defined by labeled protein expression within 2-fold of endogenous Gα_s detected in wild type cells, and we further verified that this level of expression functionally rescues the characteristic Iso-induced cytoplasmic cAMP response (Supplementary Fig. 6).

This near-endogenous level of EGFP-Gα_s expression was detectable by confocal fluorescence microscopy but quite dim (Supplementary Fig. 6c), so we used total internal reflection fluorescence (TIRF) microscopy as a more sensitive and quantitative method to assess EGFP-Gα_s redistribution. We detected intracellular redistribution of EGFP-Gα_s by loss of fluorescence intensity from the evanescent field that selectively illuminates the basal plasma membrane relative to the cell interior. Using this approach, we observed a significant reduction of surface-localized EGFP-Gα_s upon application of Iso but not vehicle. The magnitude of this effect was much lower than observed in β2AR-overexpressing cells (Fig. 3 and Supplementary Fig. 7), consistent with a considerably lower level of endogenous β2AR expression. VIP and NECA also triggered a comparable amount of intracellular redistribution of EGFP-Gα_s through their endogenous receptors (Fig. 3 and Supplementary Fig. 7). Together, these results indicate that a variety of G_s-coupled GPCRs share the ability to trigger intracellular redistribution of Gα_s under conditions of native or near-native expression.

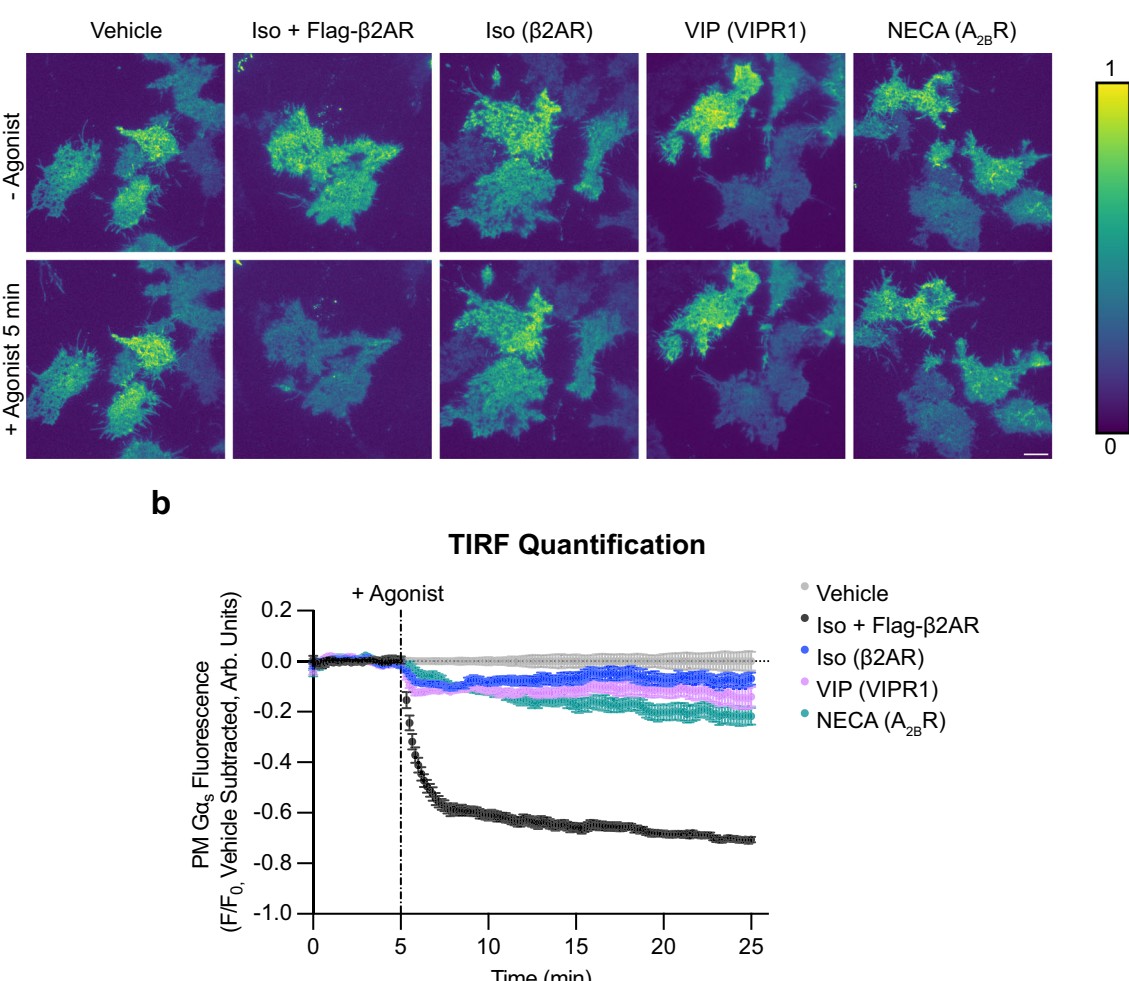

**Fig. 3 | Multiple Gₛ-coupled GPCRs drive Gαₛ redistribution at native levels.**
**a** Representative stills from time-lapse TIRF microscopy of Gαₛ KO1 + EGFP-Gαₛ rescue cells stably expressing EGFP-Gαₛ before or after 5 min of agonist treatment to activate endogenously expressed GPCRs (β2AR, VIPR1, or A$_{2B}$R). Images are shown as heat maps normalized to t = 3 min before drug addition for each individual movie. Cells were treated with either vehicle, Iso (1 μM, ± overexpression of Flag-β2AR), VIP (1 μM), or NECA (20 μM). Scale bar = 10 μm. **b** Quantification of Gαₛ fluorescence from TIRF movies depicted in (**a**). The average of vehicle control movies (n = 5) at each time point was subtracted before plotting data. Data are represented as mean ± S.E.M. of individual movies (n = 4 (Iso + Flag-β2AR, Iso, NECA) or 5 (vehicle, VIP) movies from at least three independent experiments (2–5 cells per movie)). Significance determined by repeated measures 2-way ANOVA with Dunnett's multiple comparisons test (see source data for p values). Source data are provided as a Source Data file.

## VIPR1 produces active-state Gαₛ and Gα$_{q/11}$ on the plasma membrane and Gαₛ on endosomes

The current understanding holds that Gₛ-mediated signaling requires Gαₛ to be in its active (GTP-bound) conformation[2,37]. The above experiments provided useful information about the overall distribution of Gαₛ but no information about its activation state. To address this, we utilized KB1691, a peptide shown previously to bind selectively to GTP-bound, active-state Gαₛ and to be a useful biosensor for detecting active-state Gαₛ in cells that overexpress both receptors and Gₛ[37,38]. We wondered if this peptide can be adapted to detect active-state Gαₛ at endogenous G protein expression levels. We focused on VIPR1 because this GPCR robustly triggers Gₛ-mediated signaling from both the plasma membrane and endosomes, resulting in sequential signaling phases that are temporally well-resolved[16]. We began by assessing VIP-stimulated recruitment of KB1691 to the plasma membrane, fusing KB1691 to SmBit-mApple (SmBit-mApple-KB1691) and measuring complementation with a plasma membrane-targeted LgBit (LgBit-CAAX, Fig. 4a). A clear VIP-induced recruitment signal was detected in cells expressing only endogenous G proteins (Fig. 4b). These results indicate that KB1691 is sufficiently sensitive to detect

membrane accumulation of active-state G proteins at native expression levels, provided that the activating GPCR is overexpressed.

To test the specificity of sensor recruitment, we used knockout cells lacking Gαₛ and additionally deleted Gα$_{olf}$, a close paralog of Gαₛ (Gα$_{s/olf}$ DKO cells) that we presumed would also engage the sensor (Supplementary Fig. 5). The VIP-induced biosensor response was markedly reduced in Gα$_{s/olf}$ DKO cells, verifying the ability of the sensor to detect active-state Gαₛ at native levels (Fig. 4b). We were surprised to observe a residual recruitment signal remaining in Gα$_{s/olf}$ DKO cells, with faster onset and shorter duration (Fig. 4b), despite the VIP-induced cAMP response being fully abolished in the same cells and functionally rescued by EGFP-Gαₛ re-expression as expected (Supplementary Fig. 5 and 6). Together, these results suggest that KB1691 detects not only active-state Gαₛ in the present experimental system, but also a kinetically distinct, non-Gₛ component that is activated by VIPR1.

As VIPR1 has been observed previously to initiate signaling through both Gₛ and G$_q$[39], we wondered if the non-Gₛ component might represent active-state Gα$_q$ and/or Gα$_{11}$. Consistent with this idea, the G$_{q/11}$ inhibitor YM-254890 (YM)[40] eliminated the residual VIP-

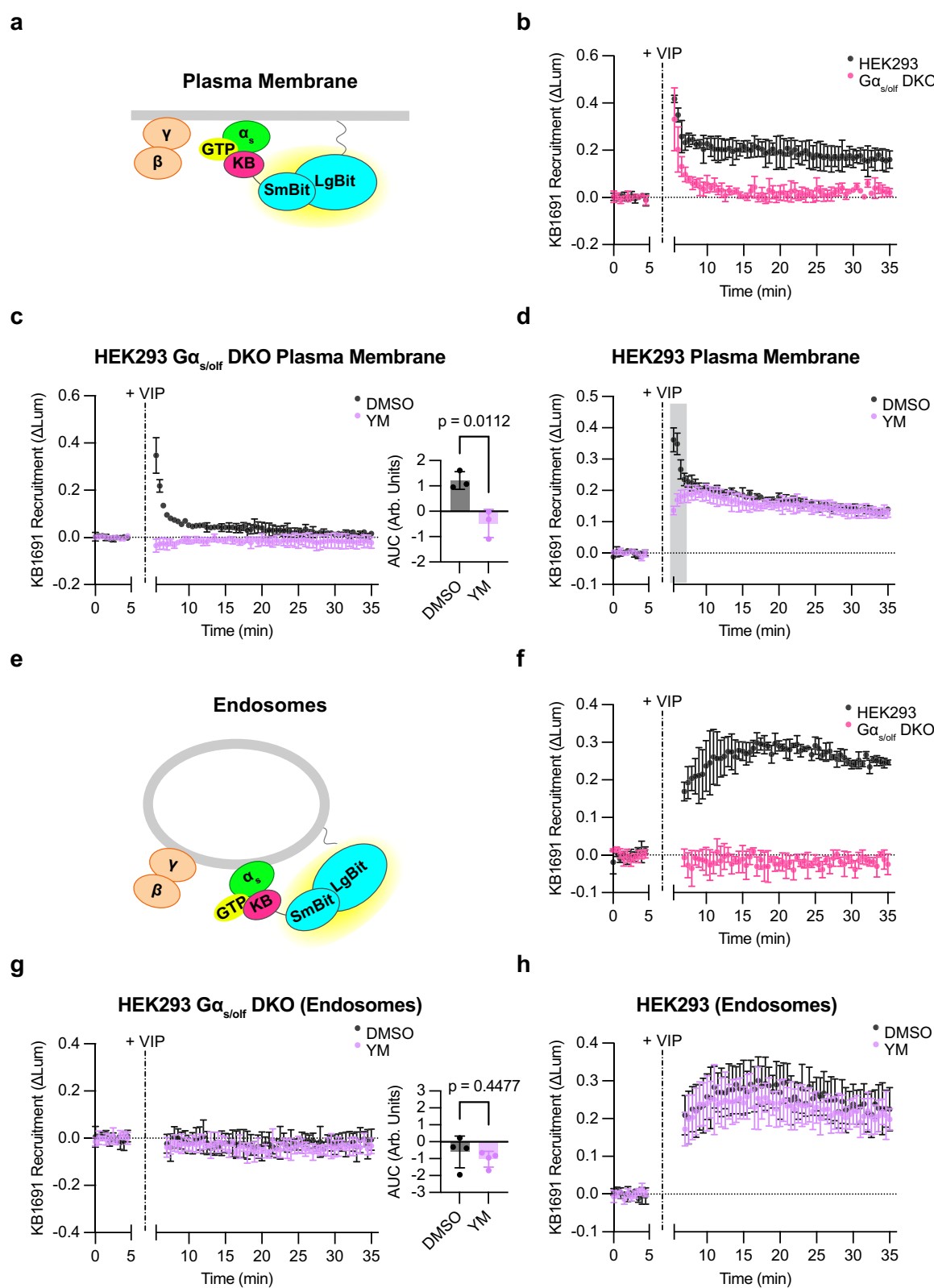

induced recruitment signal in Gα$_{s/olf}$ DKO cells (Fig. 4c). Furthermore, the residual, faster component detected by KB1691 was absent in Gα$_{q/11}$ DKO cells (Supplementary Fig. 8a)[41]. Moreover, in contrast to parental cells, YM had no effect on KB1691 recruitment in Gα$_{q/11}$ DKO cells (Supplementary Fig. 8a). Together, these data suggest that the non-G$_s$ recruitment signal reflects active-state Gα$_{q/11}$. To further test this interpretation, we adapted the Gα$_{q/11}$ binding domain of p63RhoGEF, a

previously described biosensor of active-state Gα$_{q/11}$[42], into our NanoBit assay platform (Supplementary Fig 8b). As predicted, VIPR1 caused detectable YM-sensitive recruitment of p63RhoGEF to the plasma membrane (Supplementary Fig 8c). In addition, the rapid peak of p63RhoGEF recruitment mirrored the kinetics of the YM-sensitive component detected by KB1691 (Fig. 4c, d, Supplementary Fig. 8c). Taken together, these data strongly suggest that the non-G$_s$ signal

**Fig. 4 | Detection of active-state, endogenous Gα subunits on both the plasma membrane and endosomes. a** Schematic of KB1691 (active-state Gα$_s$ biosensor) plasma membrane NanoBit bystander assay. **b** NanoBit bystander assay showing recruitment of KB1691 to the plasma membrane in both HEK293 parental cells and Gα$_{s/olf}$ DKO cells expressing Halo-VIPR1. VIP (1 μM) was added after 5 min. **c** Left: NanoBit bystander assay showing recruitment of KB1691 to the plasma membrane in Gα$_{s/olf}$ DKO cells expressing Halo-VIPR1 and pretreated with either DMSO (0.1 %) or YM-254890 (1 μM, 30 min). Right: AUC of time course. VIP (1 μM) was added after 5 min. Significance determined by two-tailed unpaired t test. **d** NanoBit bystander assay showing recruitment of KB1691 to the plasma membrane in HEK293 cells expressing Halo-VIPR1 and pretreated with either DMSO (0.1 %) or YM-254890 (1 μM, 30 min). VIP (1 μM) was added after 5 min. Shaded areas represent time points at which the difference between DMSO- and YM- treated cells are statistically significant ($p < 0.05$, determined by repeated measures 2-way ANOVA with Sidak's

multiple comparisons test, see source data). **e** Schematic of KB1691 endosome NanoBit bystander assay. **f** NanoBit bystander assay showing recruitment of KB1691 to endosomes in both HEK293 parental cells and Gα$_{s/olf}$ DKO cells expressing Halo-VIPR1. VIP (1 μM) was added after 5 min. **g** Left: NanoBit bystander assay showing recruitment of KB1691 to endosomes in Gα$_{s/olf}$ DKO cells expressing Halo-VIPR1 and pretreated with either DMSO (0.1 %) or YM-254890 (1 μM, 30 min). Right: AUC of time course. VIP (1 μM) was added after 5 min. Significance determined by two-tailed unpaired t test. **h** NanoBit bystander assay showing recruitment of KB1691 to endosomes in HEK293 cells expressing Halo-VIPR1 and pretreated with either DMSO (0.1 %) or YM-254890 (1 μM, 30 minutes). VIP (1 μM) was added after 5 min. Significance (n.s., see source data) determined by repeated measures 2-way ANOVA with Sidak's multiple comparisons test. Data are shown as mean ± S.D. of 3 (**b,c,d,f**) or 4 (**g,h**) independent experiments. Source data are provided as a Source Data file.

produced by VIPR1 activation indeed represents active-state Gα$_{q/11}$. We therefore defined the biosensor signal measured in the presence of YM as active-state Gα$_s$ and the YM-sensitive component as active-state Gα$_{q/11}$, and we conclude that VIPR1 stimulates the accumulation of both active-state Gα$_s$ and active-state Gα$_{q/11}$ on the plasma membrane in this system (Fig. 4d, Supplementary Fig. 8a).

We next asked if active-state G proteins can be detected also at endogenous levels on the endosome membrane. To address this, we modified the assay to measure complementation of SmBit-mApple-KB1691 with an endosome-targeted LgBit construct (endofin-LgBit, Fig. 4e). VIP indeed produced a robust endosomal recruitment signal (Fig. 4f). The kinetics of SmBit-mApple-KB1691 recruitment to endosomes were clearly slower than to the plasma membrane, as the endosome signal reached its maximum after 15 to 20 minutes of agonist application (Fig. 4f) while the plasma membrane signal peaked within several minutes (Fig. 4b). Interestingly, endosomal recruitment of the KB1691-derived sensor was abolished in Gα$_{s/olf}$ DKO cells (Fig. 4f) and was unaffected by either YM treatment (Fig. 4g, h) or Gα$_{q/11}$ knockout (Supplementary Fig. 8d). These observations indicate that VIPR1-triggered accumulation of active-state Gα indeed occurs at native levels on endosomes, as well as the plasma membrane. However, there appears to be a type of location bias in active-state Gα accumulation between these membranes, as defined by VIPR1 increasing both active-state Gα$_s$ and Gα$_{q/11}$ on the plasma membrane but selectively active-state Gα$_s$ on endosomes.

### Active-state Gα$_s$ production on endosomes is endocytosis-dependent

We next asked if the production of active-state Gα$_s$ on endosomes depends on the presence of activated GPCRs in the endosome membrane. This is expected based on the present model that G$_s$ activation at endosomes specifically requires a second GPCR-G protein coupling reaction occurring locally at the endosome limiting membrane[4–6]. However, as our results (Fig. 2, Supplementary Fig. 3) and the results of others[21,23] indicate that intracellular redistribution of Gα$_s$ can be efficiently triggered by GPCRs in the absence of receptor internalization, we sought to explicitly determine if endosomal production of active-state Gα$_s$ indeed depends on the presence of active-conformation receptors in endosomes.

To address this question, we inhibited VIPR1 internalization by viral expression of dominant negative (K44E) mutant dynamin, as described previously[16], and verified strong endocytic inhibition by fluorescence flow cytometry (Supplementary Fig. 9a, b). Mutant dynamin reduced the amount of active-conformation VIPR1 at endosomes after VIP application, as detected using a previously described active-conformation GPCR sensor (miniGs)[16], and it increased the amount of activated VIPR1 remaining at the plasma membrane (Fig. 5a–c). This confirms the ability of endocytic inhibition to reduce the accumulation of activated VIPR1 in endosomes. Mutant dynamin did not fully block the endosomal activation Gα signal (Fig. 5c), however,

suggesting that internalization of only a small fraction of the total receptor pool is sufficient to produce a robust activation signal on endosomes. Nevertheless, mutant dynamin decreased the accumulation of active-state Gα subunits on endosomes to a comparable degree as it decreased the measured accumulation of active-conformation receptors in the endosome membrane (Fig. 5d–f, Supplementary Fig. 9c, d). Together, these observations suggest that the production of active-state Gα$_s$ on endosomes, while apparently a highly sensitive process, is dependent on GPCR internalization and the presence of activated receptors in the endosome membrane.

### Detection of VIPR1-mediated coupling to Gα$_s$ on endosomes

We next sought to localize the coupling reaction that produces active-state Gα$_s$. To do so, we adapted the NanoBit assay to detect recruitment of Nb37 rather than KB1691. Nb37 is a nanobody that binds to the α-helical domain of Gα$_s$ when Gα$_s$ is stabilized by the activated receptor in a nucleotide-free state, corresponding to the catalytic intermediate thought to underlie physiological GPCR-G$_s$ coupling[43]. Thus, we interpret Nb37 binding as a biosensor of GPCR-G protein coupling[10], in contrast to KB1691, which we interpret as a biosensor of the GTP-bound, active-state G protein that is produced by the coupling reaction[37]. Nb37 was shown previously to detect coupling on endosomes in cells overexpressing both VIPR1 and G$_s$[16] and we found it possible to detect coupling in cells expressing native G proteins using the same experimental design, provided that VIPR1 was overexpressed (Fig. 6a–c). VIP produced a robust Nb37 recruitment signal on both the plasma membrane and endosomes, and with similar sequential kinetics as the KB1691 recruitment signal (Fig. 6c). We conclude that local GPCR-G$_s$ coupling indeed occurs on endosomes, and that it likely occurs continuously during prolonged agonist exposure to sustain active-state Gα$_s$ production.

We were surprised to also observe a residual Nb37 recruitment signal remaining in Gα$_{s/olf}$ DKO cells. Similar to results obtained with KB1691 (Supplementary Fig. 10a), this non-G$_s$ signal was eliminated in the presence of YM and absent in Gα$_{q/11}$ DKO cells (Supplementary Fig. 10b–d). Accordingly, we defined separate G$_s$ and G$_{q/11}$ components of Nb37 recruitment according to YM-sensitivity, as with KB1691. Using this approach, we resolved with Nb37 both G$_s$ and G$_{q/11}$ components of VIPR1-mediated coupling on the plasma membrane, but primarily a G$_s$ component on endosomes (Fig. 6c, Supplementary Fig. 10c–f). These results further support the existence of location bias in G protein activation, based on VIPR1 coupling to both G$_s$ and G$_{q/11}$ on the plasma membrane but primarily to G$_s$ on endosomes.

### The location bias of GPCR-G protein coupling on endosomes is GPCR-specific

As an additional approach to test the idea that endosomal G$_s$ activation requires local coupling, we took advantage of the fact that the human A$_{2B}$R naturally internalizes very weakly after agonist-induced activation when compared to VIPR1[34]. If local coupling to activated receptors

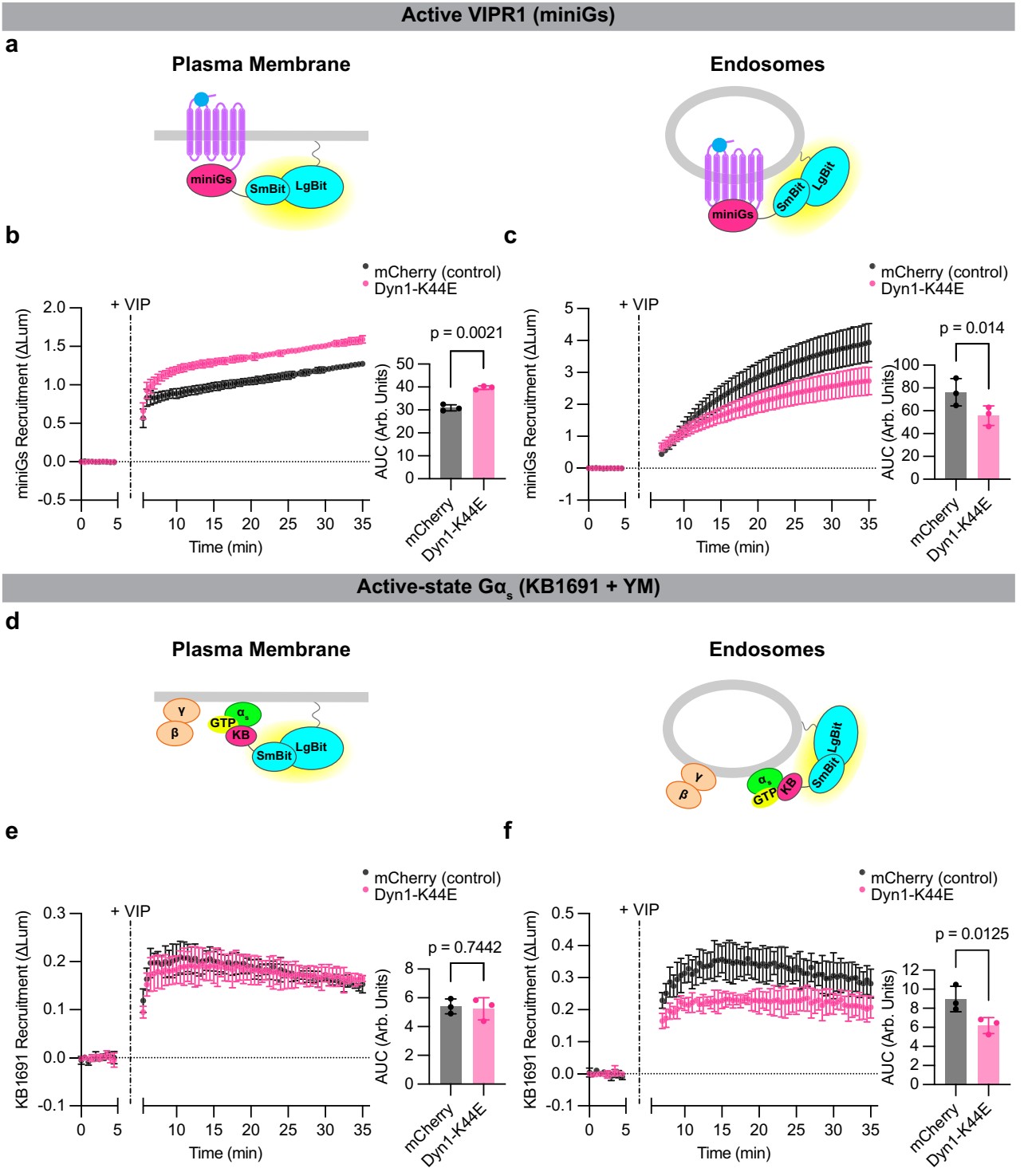

**Fig. 5 | Active-state Gα_s production on endosomes depends on VIPR1 endocytosis. a** Schematics of miniGs (active receptor biosensor) plasma membrane (left) or endosome (right) NanoBit bystander assays. **b,c** NanoBit bystander assays showing recruitment of miniGs to the plasma membrane (**b**) or endosomes (**c**) after activation of Halo-VIPR1 in HEK293 cells co-expressing mCherry (control) or mCherry-Dyn1-K44E. Left: time course; Right: AUC quantification. VIP (1 μM) was added after 5 min. Significance determined by paired two-tailed t-test. **d** Schematics of KB1691 (active-state Gα_s biosensor) plasma membrane (left) or endosome (right) NanoBit bystander assays. **e,f** Left: NanoBit bystander assay showing Gα_s-specific

recruitment of KB1691 to the plasma membrane (**e**) or endosomes (**f**) after activation of Halo-VIPR1 in HEK293 cells co-expressing mCherry (control) or mCherry-Dyn1-K44E and pretreated with YM-254890 (1 μM, 30 min). Right: AUC of time course. VIP (1 μM) was added after 5 min. Data for DMSO-treated control cells are shown in Supplementary Fig. 9. Significance was determined by repeated measures 2-way ANOVA with Sidak's multiple comparisons test with DMSO control data shown in Supplementary Fig. 9 (see source data). Data are shown as mean ± S.D. of 3 independent experiments. Source data are provided as a Source Data file.

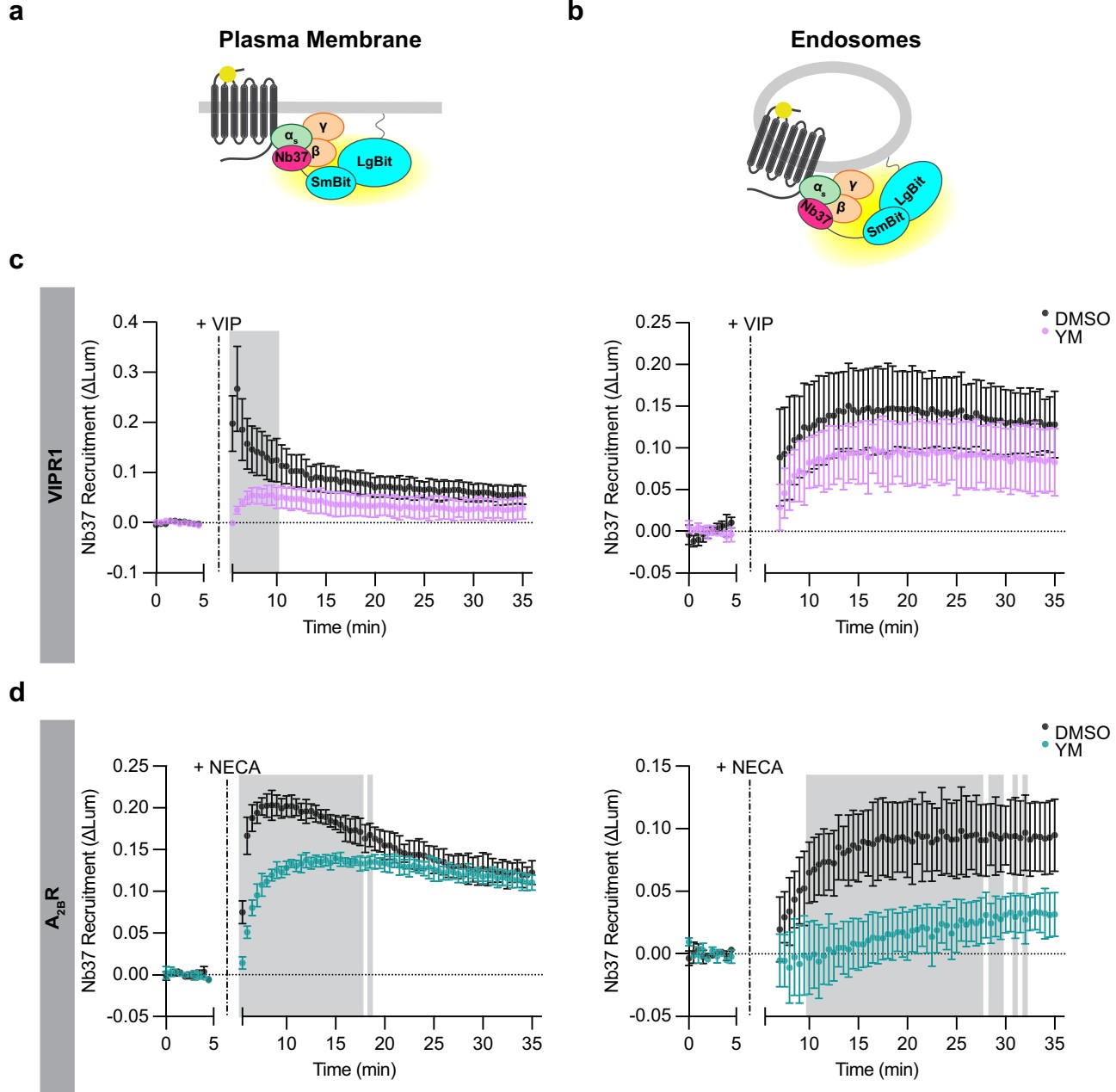

**Fig. 6 | Differential location bias in endosomal G protein coupling by VIPR1 and A$_{2B}$R. a,b** Schematics of Nb37 (GPCR-G protein coupling biosensor) plasma membrane (**a**) and endosome (**b**) NanoBit bystander assays. **c** NanoBit bystander assays depicting Nb37 recruitment to the plasma membrane (left) or endosomes (right) after Halo-VIPR1 activation in HEK293 cells pretreated with either DMSO or YM-254890 (1 μM, 30 min). Cells were treated with VIP (1 μM) at 5 min. **d** NanoBit bystander assays depicting Nb37 recruitment to the plasma membrane (left) or endosomes (right) after Halo-A$_{2B}$R activation in HEK293 cells pretreated with either DMSO or YM-254890 (1 μM, 30 min). Cells were treated with NECA (100 μM) at 5 min. Shaded areas in (**c,d**) represent time points at which the difference between DMSO- and YM- treated cells are statistically significant ($p < 0.05$, determined by repeated measures 2-way ANOVA with Sidak's multiple comparisons test, see source data). Data are shown as mean ± S.D. of 3 (**c**, left, and **d**) or 4 (**c**, right) independent experiments. Source data are provided as a Source Data file.

is required for endosomal G protein activation to occur, we predicted that A$_{2B}$R would produce a relatively weak G$_s$ coupling signal on endosomes. This was indeed the case. A$_{2B}$R produced detectable coupling to G$_s$ and G$_{q/11}$ at the plasma membrane, as indicated by both YM-insensitive and -sensitive components of Nb37 recruitment (Fig. 6d and Supplementary Fig. 10g, h), but little or no coupling to G$_s$ at endosomes (Fig. 6d).

A$_{2B}$R did produce a YM-sensitive component of Nb37 recruitment to endosomes, however, suggesting local activation selectively of G$_{q/11}$ (Fig. 6d). This was unexpected, in light of A$_{2B}$R internalizing poorly after activation by NECA, but it supports and extends the concept of

location bias in endosomal relative to plasma membrane G protein activation. Specifically, our results suggest that VIPR1 and A$_{2B}$R are similar in their ability to activate both G$_s$ and G$_{q/11}$ at the plasma membrane but, at endosomes, VIPR1 preferentially activates G$_s$ while A$_{2B}$R activates G$_{q/11}$. Accordingly, distinct GPCR family members appear to differentially bias local G protein activation on endosomes in a receptor-specific manner (Fig. 7).

## Discussion

Many GPCRs are now known to exist in an activated conformation on both endomembranes and the plasma membrane[4–6], and there is

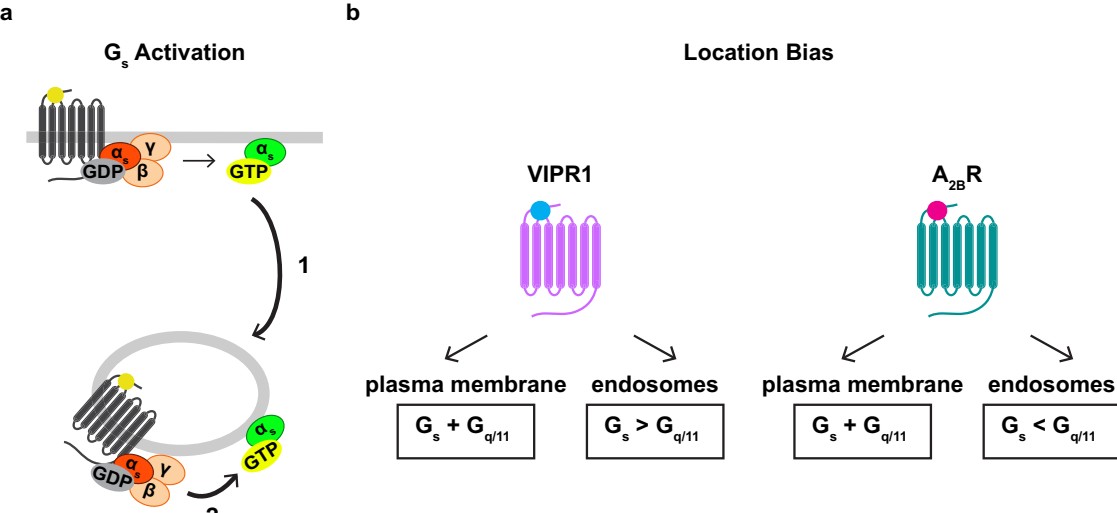

**Fig. 7 | Proposed models of regulation of endosomal G protein activity.**
**a** Discrete steps of Gα_s translocation and activation assessed in this study. Coupling of $G_s$ to an activated GPCR at the plasma membrane activates $G_s$ and promotes Gα_s redistribution to intracellular membranes, including endosomes (1). A second GPCR-$G_s$ coupling reaction on endosomes promotes the accumulation of active-state Gα_s on the endosome limiting membrane (2). **b** Model of location bias in G protein activation by VIPR1 and $A_{2B}R$. VIPR1 activates both $G_s$ and $G_{q/11}$ on the plasma membrane but preferentially activates $G_s$ on endosomes. In contrast, $A_{2B}R$ activates both $G_s$ and $G_{q/11}$ at the plasma membrane but preferentially activates $G_{q/11}$ on endosomes.

accumulating evidence that they produce distinct and additional effects from endomembranes[9,34,44–53]. Such signaling requires GPCRs to increase G protein activity on the appropriate endomembrane compartment, but how this is achieved remains unclear. We investigated this question by focusing on how $G_s$-coupled receptors regulate G protein localization to, and activity on, endosomes.

The ability of G protein activity to redistribute between membranes was first proposed more than 40 years ago based on in vitro biochemical reconstitution[54]. Multiple groups have since demonstrated redistribution of Gα_s in intact cells and provided insight into its mechanistic basis[20–25]. Here, we began by verifying the present view that 1) $G_s$-coupled GPCRs trigger the rapid redistribution of Gα_s from the plasma membrane to sample various intracellular membranes, including endosomes[20,23–25], 2) this process is reversible[20,21], and 3) it does not require the triggering GPCR to internalize[21,23]. We did so by focusing on the β2AR as the $G_s$-coupled GPCR prototype studied previously to interrogate Gα_s redistribution[20,21,23–25]. We then extended this understanding by showing that a variety of other $G_s$-coupled GPCRs also trigger a similar intracellular redistribution of Gα_s when expressed at native levels. We next carried out a series of experiments using conformational biosensors that reveal location-specific regulation, both of the active-state Gα subunit accumulation (KB1691) and of the coupling reaction that produces active-state Gα subunits (Nb37). In addition, we demonstrate that this biosensor approach can detect these key early signaling steps at native G protein levels.

Our results support a simple model in which the localization and activation of Gα_s on endosomes are separately regulated by distinct GPCR-G protein coupling reactions that occur at different subcellular locations, with coupling on the plasma membrane increasing endosomal Gα_s localization and local coupling on endosomes increasing endosomal Gα_s activity (Fig. 7a). However, we observe a small amount of Gα_s on endosomes in cells prior to agonist exposure, as noted previously by others[20,28]. Thus, it remains to be determined if the first coupling reaction, which increases Gα_s association with endosomes, is essential for signaling from endosomes, or if the basal level of endosomal Gα_s association is sufficient. We note that previous studies have come to different conclusions on this question, albeit based on studies of different G protein classes[19,20], and that significant aspects of Gα_s trafficking and its relation to endosomal signaling remain

unresolved[20,28,55]. Further, we focused here on only three GPCRs; therefore, the completeness and generality of our conclusions remain to be more fully investigated. For example, while our results suggest that Gα_s traffics separately from receptors, a recent study reported that vasopressin-2 receptor internalization drives endosomal localization of the $G_s$ heterotrimer[55]. In addition, as we found that even strong endocytic inhibition did not fully inhibit the accumulation of active-state Gα_s on endosomes (Fig. 5c), it remains possible that other factors, such as non-receptor GEFs[56], influence the measured endosomal G protein activity.

The present results provide insight into the detection and localization of active-state, GTP-bound Gα_s in intact cells that express only native G proteins. In evaluating the specificity of the KB1691 biosensor used to achieve this, we were surprised to observe a distinct non-$G_s$ component, defined by its different kinetics and presence in Gα_{s/olf} DKO cells. Based on our experiments using pharmacological and genetic manipulations, we interpret this component as active-state Gα_{q/11}. We observed similar results with the Nb37 biosensor, which we interpret as a probe of the GPCR-G protein coupling reaction. Here again, we observed a non-$G_s$ component that is consistent with Nb37 detecting GPCR coupling to $G_{q/11}$ as well as $G_s$. We found that both VIPR1 and $A_{2B}R$ produce $G_s$ and $G_{q/11}$ activation components at the plasma membrane, but that they preferentially produce one component or the other on endosomes in a receptor-specific manner (Fig. 7b). These results support the existence of location bias in GPCR-G protein coupling selectivity on endomembranes, consistent with previous evidence for such bias both through assays of functional signaling[8,57] and recruitment of GPCR activation sensors[58–60]. In addition, they suggest that this location bias is GPCR-specific. How such receptor-specific location bias is achieved, and whether or how this selectivity influences downstream functional responses triggered by distinct GPCRs under physiological conditions, remain intriguing questions for future study.

It is also remarkable that $A_{2B}R$ produced a detectable $G_{q/11}$ coupling signal on endosomes because this GPCR internalizes very weakly when compared to VIPR1[34]. This might indicate the existence of a distinct endosomal G protein activation mechanism, as we have described recently for $G_{i/o}$ activation on endosomes[61] and others have suggested may occur for $G_{q/11}$[19]. However, because the G protein activation signals that we detect with endogenous G proteins require

significant overexpression of the activating GPCR, it is possible that the activation detected on endosomes requires the presence of only a small fraction of activated receptors in the endosome membrane. It also remains possible that the endosomal activation signals observed reflect functional crosstalk between $G_s$ and $G_{q/11}$ pathways, as has been observed by others with different GPCRs[62]. Clearly, much remains to be learned about the subcellular organization of G protein activation and its location-specific regulation.

In closing, the present results add to the currently expanding mechanistic framework of spatiotemporal GPCR signaling through heterotrimeric G proteins. They also raise new questions that may help to guide further elucidation of this process and enable its future therapeutic manipulation.

## Methods

### Cell culture and transfections

HEK293 cells were purchased from ATCC (CRL-1573) and cultured in DMEM (Gibco 11965-092) and 10 % FBS (R&D Systems, S12495) at 37 °C and 5 % $CO_2$ in a humid environment. All cell lines used in this study were generated from HEK293 cells, with the exception of the HEK293A parental cells used in Supplementary Figs. 8 and 10[41], the HEK293a-derived *GNAS/GNAL* double knockout[63] cells used in Supplementary Fig. 5, and the HEK293-derived *GNAQ/GNA11* double knockout cells[41] used in Supplementary Figs. 8 and 10. Polyclonal cells stably expressing Flag-β2AR[34] were cultured in 500 µg/mL geneticin (Gibco 10131027), and VIPR1 knockout cells stably expressing tet-inducible (TRE3G) Halo-VIPR1[41] were cultured in 2 µg/mL puromycin. All cell lines were routinely screened for mycoplasma contamination (MycoAlert, Lonza LT07-318). Cells were transfected using Lipofectamine 2000 (Thermo Fisher 11668019) according to the manufacturer's protocol. For experiments in Fig. 5 and Supplementary Fig. 9c, d, cells were transfected with appropriate receptor and nanobit pairs and transduced after four hours with either mCherry or mCherry-Dynamin-1 K44E BacMam diluted in culture media.

### DNA constructs and molecular cloning

All DNA constructs used in this study are listed in Supplementary Table 1. Novel constructs were constructed by standard InFusion (Takara Bio) or KLD (NEB) cloning techniques, following the manufacturers' protocols, and sequences were confirmed by Sanger sequencing. mCherry and mCherry-Dynamin-1 K44E BacMam produced from pCMV-Dest (Thermo Fisher A24223) used in Fig. 5 and Supplementary Fig. 9 were produced according to the manufacturer's protocol.

### Generation of CRISPR KO cell lines

Single guide RNAs (sgRNAs) were designed with the Synthego CRISPR design tool (Supplementary Table 2). To generate ribonucleoproteins (RNPs), 3 µL of 53.3 µM sgRNA (Synthego) were mixed with 2 µL of 40 µM Cas9 (UC Berkeley Macrolab) and incubated at room temperature for 10 min. Cells ($2.0 \times 10^5$) were prepared for electroporation with RNPs with the SF Cell Line 4D Nucleofector kit (Lonza) following the manufacturer's protocol and electroporated in a 4D Nucleofector (Lonza) using program CM-130. After electroporation, monoclonal cell lines were established using standard techniques and genetic modifications were verified using either Sanger sequencing or next-generation sequencing (Amplicon-EZ, Azenta Life Sciences, see Supplementary Table 3 for NGS primers). For novel Gα$_{s/olf}$ DKO cells, modifications were done sequentially.

### Generation of EGFP-Gα$_s$ KO cell lines

Gα$_s$ KO1 and KO2 cells were transfected with EGFP-Gα$_s$ and selected with 500 µg/mL geneticin. After selection, cells expressing EGFP-Gα$_s$ were sorted into a polyclonal population using a FACSAria Fusion Flow Cytometer (BD Biosciences, KO1) or FACSAria III flow cytometer (BD Biosciences, KO2). Cells were cultured under continued selection.

### Live cell confocal microscopy

Confocal imaging was performed using a fully automated Nikon Ti inverted microscope equipped with a CSU-22 spinning disk (Yokogawa), piezo stage (Mad City Labs), 4-line Coherent OBIS laser launch (100 mW at 405, 488, 561, and 640 nM), a quad dichroic 405/491/561/640 (Yokogawa), and corresponding emission filters ET460/50 m, ET525/50 m, ET610/60 m, ET700/75 m in a filter wheel controlled by a Lambda 10-3B (Sutter) for channels DAPI/GFP/RFP/Cy5, respectively. Images were captured using an Apo TIRF 100x/1.49 oil objective lens (Nikon) and a Photometrics Evolve Delta EMCCD Camera (154 nm/pixel) controlled with Nikon NIS Elements HC v.5.21.03 software.

For live imaging, cells grown in either 6-well plates or 6 cm dishes were transfected 48 hours before imaging and plated into 35 mm glass bottom microscopy dishes (Cellvis D35-20-1.5-N) coated with 0.001 % (w/v) poly-L-lysine (Millipore Sigma P8920) 24 hours after transfection. Receptors were surface labeled with either monoclonal anti-Flag M1 antibody (Millipore Sigma F3040) labeled with Alexa Fluor 647 (Thermo Fisher A20186) or 200 nM JF$_{635}$i-HTL[64] for 10 min at 37 °C and 5 % $CO_2$. After labeling, cells were washed three times and imaged in imaging media (DMEM (no phenol red, Gibco 31053-028) supplemented with 30 mM HEPES pH 7.4) in a temperature- and humidity-controlled chamber at 37 °C (OkoLab). Time-lapse images in Fig. 1, Supplementary Fig. 1, Supplementary Fig. 3, and Supplementary Fig. 4 were acquired by imaging cells at 20 s intervals for 30 min, and agonist (specified in figure legends) was added after 5 min. In Supplementary Fig. 3a, cells were treated with either Dyngo4a (30 µM, Abcam ab120689) or DMSO (0.1 %) for 25 min prior to imaging, Alexa Fluor 647 coupled anti-Flag M1 antibody was added for the last 10 minutes of pretreatment, and cells were washed and imaged in imaging media containing either Dyngo4a or DMSO. In Supplementary Fig. 4b, cells were co-treated with 10 µM H89 (Cell Signaling Technology 9844) and 200 nM JF$_{635}$i-HTL[64] (to surface label the receptors) for 10 min at 37 °C and 5 % $CO_2$, and then washed and imaged in imaging media supplemented with 10 µM H89. For time lapse images in Fig. 2 and Supplementary Fig. 2, cells were imaged at 20 s intervals for 60 min with 100 nM Iso added at 5 min and 10 µM Alp added at 15 min.

Images were processed for presentation in Fiji v2.14[65]. Pearson correlation analysis was performed using Cell Profiler v4.2.6[66]; briefly, cells were segmented based on the green channel for each frame and Pearson correlation was calculated between channels at each time point. Line scan analysis was performed in Fiji v2.14[65]; briefly, the fluorescence intensity along the indicated regions was measured and normalized to the average fluorescence intensity along the line and then plotted in GraphPad Prism.

### Fixed imaging

Cells were transfected in 6-well plates 48 h before fixation. After 24 hours, cells were split onto coverslips coated with 0.001 % poly-L-lysine (Millipore Sigma P8920) in 12-well plates and grown for an additional 24 h. Cells were then surface labeled with monoclonal anti-Flag M1 antibody (Millipore Sigma F3040) labeled with Alexa Fluor 647 (Thermo Fisher A20186) for 10 min at 37 °C and 5 % $CO_2$. After labeling, cells were washed two times with full media (DMEM + 10 % FBS) and treated with either vehicle or Iso (1 µM) for an additional 15 min at 37 °C and 5 % $CO_2$. Cells were then placed on ice, washed 1x with DPBS, and fixed at room temperature for 10 min in 3.7 % formaldehyde in modified BRB80 (80 mM PIPES pH 6.8, 1 mM $MgCl_2$, 1 mM $CaCl_2$). After fixation, cells were washed three times with DPBS, incubated in TBS for 20 min, and washed an additional three times with DPBS. Dapi (1:5000) was included in the final wash. Cells were then mounted in ProLong Gold Antifade mounting medium and left to dry overnight in the dark. Slides were imaged using the confocal microscope described above using a Plan Apo VC 60x/1.4 oil objective lens. Images were processed for presentation using Fiji v2.14[65]. Line scan analysis was performed in Fiji v2.14[65]; briefly, the fluorescence intensity along the indicated

regions was measured and normalized to the average fluorescence intensity along the line and then plotted in GraphPad Prism.

## TIRF microscopy

For live cell TIRF microscopy, cells were imaged using the same methods as for live cell confocal microscopy. Images were acquired on a fully automated inverted Nikon Ti-E microscope controlled by Nikon NIS-Elements software (5.20.00 build 1423), a Nikon motorized stage equipped with a TIRF module with STORM lens (Nikon), Agilent MLC400 (405, 488, 561, 647 nm) light source with NIDAQ interface (v18.00), and corresponding emission filters ET455/50 m, ET525/50 m, ET600/60 m, ET705/72 m in a filter wheel controlled by a Lambda 10-3B (Sutter) for channels DAPI/GFP/RFP/Cy5, respectively. Images were captured using an Apo TIRF 100x/1.49 objective (Nikon) with an Andor DU897 EMCCD camera and an OkoLab temperature controlled live stage insert. Time lapse images were acquired at 10 s intervals for 25 min at 37 °C with agonist added at 5 min.

Images were analyzed and processed for presentation using Fiji v2.14[65]. To quantify relative changes in surface fluorescence ($F/F_0$), cells were manually segmented by drawing a region of interest (ROI) around the cell surface. The mean, background subtracted, fluorescence intensity (F) was measured at each time point and normalized to the average mean fluorescence intensity before agonist treatment ($F_0$). $F/F_0$ values for each cell within an individual movie ($n = 2$–6 cells per movie) were then averaged to calculate average $F/F_0$ values for each individual movie. To calculate vehicle subtracted $F/F_0$ values, the $F/F_0$ values of vehicle control movies were averaged at each time point and subtracted from the $F/F_0$ values at each time point for individual movies. For presentation in Fig. 3 and Supplementary Fig. 7, images were scaled to the pre-agonist time point and pseudo-colored using the viridis colormap in Fiji to visualize relative changes in fluorescence.

## NanoBit luciferase complementation assays

Cells were grown in 6-well plates or 6 cm dishes and transfected with both receptor constructs (Flag-β2AR, Halo-VIPR1, or Halo-A$_{2B}$R) and the appropriate LgBit- and SmBit-tagged constructs (in Nb37 assays shown in Fig. 6 and Supplementary Fig. 10, the SmBit(101) tag was used, while the SmBit(114) tag was used in all other assays[67]). After 24 h, cells were washed, lifted, spun at 500 x g for 3 min and resuspended in assay buffer (20 mM HEPES pH 7.4, 135 mM NaCl, 5 mM KCL, 0.4 mM MgCl$_2$, 1.8 mM CaCl$_2$) with 5 μM coelenterazine-H (Research Product International C61500). Cells were plated (100 μL) into untreated white 96-well plates (Corning 3912) and incubated at 37 °C and 5 % CO$_2$ for either 10 or 30 min (for cells pretreated with DMSO (0.1 %) or YM-254890 (1 μM, Cayman Chemical 29735 or Tocris 7352), as indicated in figure legends) before reading. Luminescence was measured on either a Synergy H4 (BioTek, for data in Fig. 2 and Supplementary Fig. 3) or Spark (Tecan, for all other data) plate reader. For assays in Fig. 2, luminescence was read every 1 minute for a 5-minute baseline, after which either vehicle or Iso (100 nM) was added and luminescence read for 10 minutes, followed by vehicle or Alp (10 μM) addition and additional luminescence reading for 30 minutes. For assays in Supplementary Fig. 3b, luminescence was read every 1 min for a 5 min baseline, after which either vehicle or Iso (1 μM) was added and luminescence read for an additional 30 min. For all other assays, luminescence was read every 30 s for a 5 min baseline, after which vehicle or agonist (noted in figure legends) was added and luminescence measured for an additional 30 min. For cells pretreated with DMSO or YM-254890, cells were kept in continual treatment for the duration of the assay.

To analyze data, the change in normalized luminescence was calculated by normalizing each well to its average baseline luminescence. Then, the average change of luminescence of vehicle-treated wells was subtracted from the average change in luminescence of agonist-treated cells. Data are represented as the vehicle subtracted change in luminescence of agonist-treated cells ($\Delta Lum = Lum_{agonist} -$

$Lum_{vehicle}$). For endosome NanoBit assays, the first three time points (60 s) after agonist addition were not shown, as a small amount (< 10 %) bleedthrough from the plasma membrane was observed intermittently, which we speculate represents a small amount of sensor mislocalization. This is not consistent with the time course of receptor internalization and does not affect the interpretation of results.

## cADDis cAMP assays

Intracellular cAMP levels were measured using either Green Up cADDis cAMP biosensor (Montana Molecular U02006, Supplementary Fig. 5) or Red Up cADDis cAMP biosensor (Montana Molecular U0200R, Supplementary Fig. 6) following the manufacturer's protocol. Briefly, 50,000 cells per well were plated into TC-treated black 96-well plates (Corning 3340) coated with 0.001 % (w/v) poly-L-lysine (Millipore Sigma P8920) and transduced with cADDis BacMam. After 24 h, cells were washed with assay buffer (20 mM HEPES pH 7.4, 135 mM NaCl, 5 mM KCL, 0.4 mM MgCl$_2$, 1.8 mM CaCl$_2$) twice and incubated at 37 °C in a temperature controlled plate reader (Tecan Spark for Green cADDis assays or BioTek Synergy H4 for Red cADDis assays). Baseline fluorescence was read with an excitation wavelength at 500 nm (Green cADDis) or 558 nm (Red cADDis) and emission wavelength at 530 nm (Green cADDis) or 603 nm (Red cADDis) for 5 min every 30 s, after which agonist (noted in figure legends) was added and fluorescence read for an additional 30 min. To calculate the change in intracellular cAMP ($\Delta F/F_0$), the average baseline fluorescence for each well was calculated ($F_0$) and the change in fluorescence for each well ($\Delta F = F - F_0$) was normalized to $F_0$.

## Internalization by flow cytometry

VIPR1 knockout cells expressing tet-inducible (TRE3G) Halo-VIPR1[16] were plated in 6-well plates and then both transduced with either mCherry control or mCherry-Dynamin 1 K44E Bacmam and induced with 1 μg/mL doxycycline overnight. After 24 h, cells were treated with 1 μM VIP for 0 or 30 min, washed two times with cold PBS-EDTA (UCSF Cell Culture Facility), and then labeled with 200 nM JF$_{635}$i-HTL[64] for 30 min at 4 °C. Cells were then washed three times with cold PBS-EDTA, lifted with TrypLE Express (Gibco 12604021), and then surface fluorescence was measured on a Cytoflex (Beckman Coulter) flow cytometer controlled by CytExpert v.1.3.1.22 (Beckman Coulter). Data were analyzed using FlowJo v.10.10.0 (B.D. Life Sciences). Populations were gated for cells expressing mCherry, and percent internalization was calculated as $((1-F_t)/F_{t0})*100$, where $F_t$ represents the median background-subtracted fluorescence intensity at time t. Each biological replicate was calculated as the average of 3 technical replicates.

## Immunoblotting

Cells were lysed in RIPA buffer (50 mM Tris pH 7.4, 150 mM NaCl, 1 % Triton X-100, 0.5 % sodium deoxycholate, 0.1 % SDS) supplemented with Roche cOmplete EDTA-free protease inhibitor tablets (Roche 04693159001) and lysate was boiled at 95 °C for 5 min in NuPage LDS Sample Buffer (Thermo Fisher, NP0007) and 20 mM DTT. For Flag-β2AR blots in Supplementary Fig. 3, lysate was incubated with LDS Sample Buffer and 20 mM DTT for 1 h at room temperature instead of boiled at 95 °C. SDS-PAGE and western blots were performed using standard techniques with polyclonal rabbit anti-Gα$_{s/olf}$ antibody (LS-Bio LS-B4790, 1:1000, blocked in Tris-buffered saline, 5 % milk, 0.1 % Tween-20), a monoclonal rabbit anti-GAPDH antibody (D16H11, Cell Signaling Technology 5174S, 1:1000, blocked in LI-COR Intercept (TBS) blocking buffer (LI-COR 927-60001)), a monoclonal rabbit βARR1/2 antibody (D24H9, Cell Signaling Technology 4674, 1:1000, blocked in LI-COR Intercept (TBS) blocking buffer), or a monoclonal mouse anti-Flag M1 antibody (Millipore Sigma F3040, 1:1000, blocked in LICOR Intercept (TBS) blocking buffer), followed by IRDye 800- or 680-linked anti-mouse or anti-rabbit IgG secondary antibodies (LI-COR Biosciences). Blots were imaged using an Odyssey Imager (v.2.0.3, LI-COR Biosciences) and quantified using Fiji (v2.14).

## Statistical analysis and reproducibility

Microscopy quantification data are presented as mean ± S.E.M. of individual dishes from at least three independent experiments, while cAMP and NanoBit data are presented as mean ± S.D. from at least three independent experiments. Each biological replicate in cAMP and NanoBit assays represents the average of at least two technical replicates. All images are representative of at least three biologically independent experiments, except for images in Supplementary Fig. 6c, which are representative of two independent experiments. Statistical tests and area under the curve calculations were performed using GraphPad Prism (v.9 and v.10).

## Reporting summary

Further information on research design is available in the Nature Portfolio Reporting Summary linked to this article.

## Data availability

Source data are provided with this paper, and all data and materials are available upon request.

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

## Acknowledgements

We thank Luke Lavis and John Janetzko for sharing reagents and advice; Aashish Manglik for sharing equipment; and Natalia Jura, Roshanak Irannejad, Barbara Panning, Rita Fagan, and the rest of the von Zastrow lab for helpful discussion and feedback on the manuscript. Imaging data for this study were acquired at the UCSF Center for Advanced Light Microscopy (Nico Stuurman, Kari Herrington, Micaela Lasser, DeLaine Larsen, and SoYeon Kim). The UCSF Helen Diller Family Comprehensive Cancer Center Laboratory for Cell Analysis (Sarah Elmes, supported by the NIH under award P30CA082103) assisted with cell line generation. This work was supported by the NIH under awards R01DA010711 and R01DA012864 (to M.v.Z.) and K99GM151441 (to E.B.). B.W. was supported by T32GM007810. A.I. was funded by KAKENHI JP24K21281 and JP25H01016 from the Japan Society for the Promotion of Science; JP22ama121038 and JP22zf0127007 from the Japan Agency for Medical Research and Development; and JPMJFR215T and JPMJMS2023 from the Japan Science and Technology Agency.

## Author contributions

B.W. and M.v.Z. conceptualized the study with assistance from E.E.B. B.W., N.M.F., E.E.B., A.N.D., and M.v.Z. designed experiments. B.W., N.M.F., A.N.D., and E.E.B. performed experiments and analyzed data. A.I. provided essential advice and reagents. B.W. and M.v.Z. wrote the manuscript with input from all authors.

## Competing interests

M.v.Z. serves on the Scientific Advisory Board of Deep Apple Therapeutics. The authors declare no other competing interests.
