## [Transparent Peer Review file · Nature Communications]

Conformational biosensors delineate endosomal G protein regulation by GPCRs

Corresponding Author: Professor Mark von Zastrow

Version 0:

Reviewer comments:

Reviewer #1

(Remarks to the Author)

In this manuscript the authors study the movement of G α s subunits and their activation on endosomes using conformational biosensors. The manuscript is divided into two parts. The first part (Figures 1-3) looks at G α s translocation to intracellular membranes with an exclusive emphasis on trafficking to endosomes. The authors show that receptor activation at the plasma membrane leads to translocation of G α s, no translocation of a G γ subunit, and increased colocalization of G α s with EEA1, a marker of early endosomes. They show that this does not depend on receptor endocytosis, is reversible and is more difficult to observe after fixation. One of the main results is that translocation can be observed after activation of endogenous receptors, although this translocation is modest and apparently doesn't result in a detectable increase in G α s on endosomes. Some of the methods used here have not been used previously to study this phenomenon, but none of the findings or concepts are new. Early studies by Insel and colleagues in S49 cells relied on endogenous adrenergic receptors and demonstrated robust redistribution of G α s (ref. 25). As the authors point out, the other key results verify findings of previous studies. Just as important, these experiments shed no new light on signaling from endosomes as the present manuscript says nothing about how the translocation phenomenon impacts signaling. Since the authors suggest that heterotrimers are required for coupling and apparently only G α s translocates it is difficult to see how translocation supports signaling.

The second part of the manuscript addresses an entirely different question and looks for active G α s on endosomes using a peptide that binds to G α s-GTP and a nanobody that binds to nucleotide-free G α s. There was virtually no doubt that this happens given prior functional studies and more direct experiments using Nb37 by the same group; the main novelty here is the use of the KB1691 peptide in an endosome assay and the detection of endogenous G α s-GTP after activation of overexpressed receptors. Differences between VIPR1 and A2BR are nice to see but are not particularly surprising given the known properties of these receptors, and Fig. 6 suggests a quantitative rather than qualitative (as implied in Fig. 7b) difference in how they activate G proteins at different locations. One of the most interesting new findings is robust signals from both G α s sensors that persist in Gs/olf DKO cells and are blocked by YM. The authors hesitate to call these signals active G α q/11, presumably because of prior studies claiming specificity of these probes, but "non-Gs" is rather unsatisfying in light of the documented specificity of the YM compound.

All in all, the manuscript contains some interesting results, but the large majority of these confirm previous studies, and the conceptual advance is rather limited.

Specific comments:

The authors claim that endosomal and plasma membrane signals show expected "sequential" kinetics but there are some anomalies that are difficult to explain. The time courses of plasma membrane and endosome active G α s signals in Fig. 5e and f, and Fig. 6c are not that different; plasma membrane signals rise for several minutes, and endosome signals are more than 50% of maximum at the first time point after VIP addition, even when endocytosis is inhibited. The time course of active receptor appearance in endosomes is much slower. "Non-Gs" signals seem to be much faster and sometimes larger for these Gs-coupled receptors. At the very least a more careful discussion of these issues is warranted.

Dynamin K44E is only partially effective in Fig. 5. Since the receptors here are overexpressed, these experiments should be supplemented with receptors that are completely defective when it comes to endocytosis, and/or additional more effective

blockers of receptor endocytosis. In any case the degree to which receptor endocytosis is blocked should be directly quantified.

A related question, why was the β adrenergic receptor ignored for the second half of the manuscript, especially since the 3S mutant is in hand?

The authors conclude that VIPR1 activates Gq/11 (non-Gs) on the plasma membrane but not on endosomes, which seems to be the case in Fig. 4f (KB1691) but less convincingly in Fig. 6c (Nb37). If Gq/11 (non-Gs) is present on endosomes (Fig. 6d), how does it avoid activation by VIPR1 at this site?

The authors conclude in the abstract that activation of endosomal G α s depends on receptor endocytosis, but in fact they only show that this activation is dependent on dynamin. Could dynamin K44E be inhibiting delivery of the peptide ligand to endosome-resident receptors? Experiments with endocytosis-deficient receptors could address this.

TIRF images show DMSO controls but there are no DMSO traces in the grouped data. Given the small magnitude of receptor-mediated translocation such controls are important in this instance.

The endosome time course traces in Figs. 4 and 5 are missing the first two time points after the addition of agonist, whereas the plasma membrane traces show data for these time points. Why?

Panels 2c and 2d are partially redundant.

Can the authors provide some explanation for why KB1691 was replaced with Nb37 for the later experiments?

The authors should make a consistent distinction between Gs heterotrimers and G α s subunits. For example, they claim in the abstract that they show that plasma membrane events control endosomal Gs abundance. This should be G α s.

“Regulate” is a better description than “control” for how plasma membrane activation affects endosome G α s abundance, as the authors do not quantify the extent to which total endosome G α s abundance changes after activation at the plasma membrane.

Reviewer #2

(Remarks to the Author)
Wysomerski et al. Comments

In this manuscript, Wysolmerski and colleagues present data to propose mechanism by which endosomal G(s) G proteins are regulated. As acknowledged by the authors, there is significant accumulated evidence that supports the presence of G proteins in endosomes and G protein activation within these intracellular compartments. Here, the authors use very elegant approaches to show that i) these G proteins “arrive” at the endosomes without independently of the activating receptor but ii) they require an active internalised receptor to be activated. These results are novel and expand our understanding of spatiotemporal GPCR signalling. The therapeutic implications of these findings remain future endeavours, but clearly such knowledge will help in that direction. Of note, during this review, a paper from the same author’s lab has been published and conceptually both papers seem related. It is suggested the authors emphasise novelty of the present work and relationship between both reports.

I have a few questions about the results presented that require consideration prior to publication.

Proposed model of Gs accumulation in endosomes – how are the authors convinced that the Gs accumulated in the endosomes is of PM origin? Is it possible that PM signalling from GPCR/Gs triggers a signal that results in redistribution of Gs from other (non-PM) compartments to accumulate in endosomes? Could the authors perform some tracking of Gs-GFP to assess this?

The authors suggest that KB1691 recruitment to endosomes, since it is not affected by YM, reflects Gs activation. As they detected a non-Gs component at the PM, one would expect then that this endosomal recruitment of KB1691 is unaffected by Gs/olf knockout?

KB1691 and Nb37 revealing a non-Gs components – as far as this reviewer understands, both KB1691 and Nb37 were originally described and demonstrated as selective for Gs G proteins. The results here seem to argue against that selectivity and propose that both these sensors can detect a “non-Gs component”. While the data seem to point in that direction, the arguments for “location bias” seem premature without the understanding of what such “non-Gs” mechanism is. With the current justification, this stands out as a major limitation of this work.

Other comments:

- While the interpretation of Fig1c is supported by the data, this would be further supported if the authors applied a mask to remove the PM region from their Pearson Correlation Coefficient calculation.
- Supp Fig2 and 3 legends have scale bars in microM rather than microm.
- As in ref 33, it would be recommended that they authors refer to the current patent on the KB1691 peptides.
- Fig4 has one panel labelled PM and another Plasma membrane – use one for consistency

Reviewer #3

(Remarks to the Author)

Review of "Conformational biosensors delineate endosomal G protein regulation by GPCRs" by Brian Wysolmerski, Emily E. Blythe, Mark von Zastrow

In their manuscript, Wysolmerski and collaborators shed light on Gs redistribution and activation at the endosome following GPCR activation in HEK293 cells. They find that stimulation of endogenous β 2AR, VIPR1 or A2BR causes a loss of Gs from the plasma membrane. In the case of β 2AR, they investigated this finding further and found that this loss is independent of receptor internalization and that it corresponds to Gs redistribution to endosomes. In the case of VIP1R and A2BR, they show that endosome-localized Gs is active and, only for VIP1R, they show that this activation is dependent on receptor internalization. Finally, they propose a location bias mechanism in G protein activation by both VIP1R and A2BR, where the proportion of Gs and non-Gs activation is both receptor- and compartment-dependent.

The results are potentially very interesting for the GPCR and cell biology community, however the manuscript suffers from the following limitations:

- The conclusions are presented as general mechanisms of GPCR functioning, but these are not supported by the data, which are receptor specific. Moreover, conclusions are based on experiments lacking sufficient number of repeats and from qualitative assessments lacking proper quantification.
- The approaches used are not so novel. Nb37, NanoBit assay, KB1691 have been already described and used in several occasions (mentioned in the text by the authors). However, Nb37 and KB1691 here are used in a clever way to distinguish between GPCR-bound G protein versus active (=GTP bound) G protein. Similarly, the NanoBit assay was adapted to these specific questions.
- I don't see the relevance of the finding that G proteins are redistributed to endosomes independently of receptor endocytosis. What does it mean physiologically speaking? What is the function of Gs redistribution to the endosome in absence of receptor?
- The finding that G proteins are active at the endosome only when the receptor is also there is somehow reassuring, but expected I would say.
- No downstream effect of location bias in G protein activation by VIPR1 and A2BR has been assessed.
- No information on how G proteins and receptors couple at the endosome is provided. Do they traffic together? Is the receptor internalized first and then the G protein recruited from the cytoplasm? Is through fusion of 2 endosomal compartments (one containing the receptor and one showing G protein presence)?
- Only G protein redistribution has been shown to happen at native levels of receptor, not G protein activation. The authors should reconcile these findings.

Apart from these concerns, please see below for other comments that should be addressed before the manuscript is suitable for publication:

1. p.3 "However, the presence of active-state G α s on endosomes has only been hypothesized." I guess the authors mean GTP-bound G protein. This needs to be clearly stated as it can be perceived that all the work done by von Zastrow's lab and others using MiniG proteins and Nb37 does not point in that direction.
2. p.3 "significantly differ in their ability to internalize after activation." Is this a finding from this study or something already known? If known, please add refs, if new... I don't see a characterization of the 3 receptors internalization in this paper, unless I missed it.
3. In Fig. 1, colocalization between G and EEA1 is missing. Fig.1d is only representative/qualitative but no quantification is shown for G α s G colocalization at the EE.
4. Also in Fig. 1 I'm surprised not to see G at the PM in unstimulated cells. I was expecting complete colocalization with Gs. Could the authors comment on this?
5. Could the experiment performed in Fig. 2 be done in a complementary way using LgBit-Gs and SmBit-endofin, to look at redistribution of Gs to and from the endosome upon Iso or Alp addition?
6. In Fig. 2e, redistribution of Gs from the endosome to the membrane upon Alp addition is observed, even in absence of receptor internalization. It would be great to hear what the authors think about this finding, if they could speculate on how Gs can sense from the endosome the presence of an antagonist acting on receptors blocked at the plasma membrane.

7. Why did the authors choose to block endocytosis of VIPR1 by overexpressing the dominant negative mutant of dynamin Dyn1-K44E instead of using Dyngo-4a as it was done for β 2AR?

8. Please add quantification for confocal images shown in Supp. Fig. 1, 2, 3 and 4. The analysis should be conducted on a bigger sample, not just 4 cells, from more than 2 independent experiments.

9. More repeats should be included for experiments that were repeated only twice. 4 cells are not sufficient. I suggest analyzing 5 cells per repeat, for a total of 15 cells.

10. There is a clear imbalance between the first part of the manuscript where all experiments are conducted using β 2AR as model GPCR and mainly live-imaging techniques, and the second part of the paper where experiments are focused on VIP1R and A2BR using the NanoBit assay. Specifically, the authors show that Gs redistribution to the endosomes is independent of receptor endocytosis only for β 2AR. It would be informative to know whether this is also true for other Gs coupled GPCRs such as VIPR1 and A2BR. Similarly, was the NanoBit assay run for β 2AR with either KB1691 or Nb37? The manuscript would benefit from adding these data. Finally, since KB1691 is conveniently tagged with mApple, it would be good to include confocal images of KB1691 localization before and after stimulation with either VIP or NECA (+ localization markers for PM and endosomes).

11. In Fig. 7a, the authors talk about a general mechanism, but they demonstrate that redistribution of Gs to endomembranes is independent of receptor endocytosis only for β 2AR. Furthermore, the representative GPCR is in purple, which in 7b represents VIPR1, for which receptor independent Gs redistribution to endosomes has not been shown. The scheme is confusing from different aspects and I recommend changing it to reflect the findings described in the manuscript.

Minor comments:

12. Since in some experiments proteins are overexpressed and in some other experiments not, it would be cleared to add on certain figure panel (e.g. Fig 1a) which tag was used, so it's clear when proteins were overexpressed (e.g. Flag-b2AR, Gs-EGFP, etc).

13. It is sometimes difficult to understand when the authors mean G protein activation or G protein-GPCR coupling. From what I read, G protein activation is monitored with KB1691, while G protein-GPCR coupling with Nb37. If that's correct, I suggest changing Supp. Figure 9 graph headers to (G protein-GPCR coupling, Nb37).

14. Supp fig. 2a please add single channel images in black and white.

15. The quality of some images is quite poor. Would it be possible to replace images in Fig. 1d +iso 20' and Supp Fig. 4b with better ones?

16. p.9 "Signaling from endosomes requires $G_{\alpha s}$ associated with the endosome membrane to be in an active state." Please add a reference to this sentence.

17. Personally, I do not like the word "abundance" used throughout the text, as to me, it implicates some quantitative meaning, while here it means purely presence of Gs. If the authors agree with my logic, could this word be changed to "presence" or "redistribution"?

Reviewer #4

(Remarks to the Author)

In this study by Wysolmerski et al., the authors reexamine previous findings related to Gs trafficking and endosomal activity and provide the first results exploring endosomal signaling with the new $G_{\alpha s}$ -GTP sensor from the Garcia-Marcos lab. These results are timely because researchers studying location bias and intracellular signaling have surely thought about adapting this biosensor to fit their research questions since its publication last year. It is certainly exciting to see that this biosensor, when adapted to measure endosomal signaling responses, can detect robust signals at endogenous G protein levels. However, the authors present some curious findings related to the specificity of two biosensors at the center of their study (KB1691 and Nb37). It feels as though they are concluding that the detection modules are not selective for $G_{\alpha s}$ while not ruling out the possibility that other types of G protein activation may affect the concentration of components on endosomal membranes. This distinction is important because the authors conclude that there is a non-Gs component being activated at endosomes for receptors like A2BR when it may be that Gs is indeed activated at endosomes, but Gq/11 activity at the plasma membrane modulates the possibility for Gs to be activated subsequently at endosomes. My suggestions can be found below:

1. YM acts as a molecular glue keeping $G_{\alpha q}$ and $G\beta\gamma$ together as an inactive heterotrimer (Mühle et al., PNAS, 2025). Can the authors rule out that YM treatment doesn't sequester $G\beta\gamma$ from $G_{\alpha s}$? Have the authors performed the same experiments in $G_{\alpha q/11}$ KO cells? In a similar study from Vilardaga and colleagues (White et al., PNAS, 2020), $G_{\alpha q/11}$ knockout cells were used and the authors observed a decrease in endosomal cAMP generation which they proposed was dependent on the

liberation of G β y and PIP3 production to promote GPCR-arrestin binding (in this instance PTHR) and internalization. In the Cell paper from Garcia-Marcos and colleagues where KB1691 was described for the first time (Janicot et al., Cell, 2024), I do not think any Gq/11 inhibitors or knockout lines were used in the context of this sensor; their experiments to confirm G protein selectivity were based on representative tagged G proteins from the four major families and only tested PTX and YM on Gi/o and Gq/11 respectively. I think that the authors from the current study need to provide more convincing evidence that this sensor is detecting something it is not, rather than the consequence of events that preceded downstream signaling at endosomes. For example, it could still be that KB1691 and Nb37 are G α s-selective, but that endosomal G α s activation is affected by Gq/11 activity at the plasma membrane – something that may be receptor-specific.

2. While the results with VIPR and DynK44E suggest that endosomal production of active state G α s is endocytosis-dependent, there could be other factors that regulate the presence of G α s-GTP at endosomes like phosphodiesterases, RGS proteins and intrinsic GTPase activity. In other words, if VIPR internalization could be blocked completely, would we still observe residual endosomal signaling and what could that be due to? Consider other receptors that have a greater dependence on arrestin/CME like V2R. Does this sensor have the dynamic range to measure the shaping of endosomal responses by other proteins like PDEs and intracellular RGS proteins?

3. Fig. 1 G γ 2 is geranylgeranylated and does not leave the membrane as readily as other gamma subunits (Masuho et al., Cell Systems, 2021). While there is a clear intracellular localization of G γ 2, the conclusion that this population of G β y supports endosomal signaling is probably isoform-specific. In fact, it has also been proposed that G β y dimers internalized with β -arrestin-bound receptors can enable a second round of G α s activation for select receptors (Sokrat et al., Communications Biology, 2024). Given these previous findings, I suggest that the authors either broaden their conclusion about how agonist-induced activation happens at endosomes or perform additional G gamma experiments with other subtypes or receptors.

4. Fig. 1C label EGFP as control or cytosolic EGFP. Same for Fig.5 and mCherry.

5. It is unclear how subpanel 2C differs from the wildtype response in panel 2D. I would suggest reorganizing the subpanels to avoid duplication of results within the same figure because it is essentially the same experiment for WT in subpanels 2C and 2D.

Version 1:

Reviewer comments:

Reviewer #1

(Remarks to the Author)

My opinion of the manuscript is essentially unchanged. The authors now more clearly assign YM-sensitive signals to activation of Gq, which is the most novel aspect of the manuscript and the only significant improvement in the revision. The main points of the revised manuscript are still that G α s subunits leave the plasma membrane and that active G α s subunits appear on endosomes. Neither of these findings is novel and no clear link between the two observations is made. Differences between two receptors do not uncover a generalizable mechanism. The authors argument about expected kinetics is not entirely convincing. The rationale for using Nb37 in most of the later experiments is also not compelling; yes of course the probes detect different things but why is this valuable since Nb37 has been used by this group for this reason before and what's important is presumably what's detected by KB1691. Altogether the conceptual advance here does not rise to the standard of Nature Communications.

Reviewer #2

(Remarks to the Author)

The authors have addressed all my comments.

Reviewer #3

(Remarks to the Author)

I thank the authors for taking into consideration my comments, providing exhaustive answers and making additions/changes to their manuscript accordingly.

I find the revised text way more explicative, thus easier to read.

Reviewer #4

(Remarks to the Author)

I am satisfied with the authors' efforts to address my comments and those of the other reviewers.

RESPONSE TO REVIEWERS

Reviewer #1 (Remarks to the Author):

In this manuscript the authors study the movement of Gas subunits and their activation on endosomes using conformational biosensors. The manuscript is divided into two parts. The first part (Figures 1-3) looks at Gas translocation to intracellular membranes with an exclusive emphasis on trafficking to endosomes. The authors show that receptor activation at the plasma membrane leads to translocation of Gas, no translocation of a G γ subunit, and increased colocalization of Gas with EEA1, a marker of early endosomes. They show that this does not depend on receptor endocytosis, is reversible and is more difficult to observe after fixation. One of the main results is that translocation can be observed after activation of endogenous receptors, although this translocation is modest and apparently doesn't result in a detectable increase in Gas on endosomes. Some of the methods used here have not been used previously to study this phenomenon, but none of the findings or concepts are new. Early studies by Insel and colleagues in S49 cells relied on endogenous adrenergic receptors and demonstrated robust redistribution of Gas (ref. 25). As the authors point out, the other key results verify findings of previous studies. Just as important, these experiments shed no new light on signaling from endosomes as the present manuscript says nothing about how the translocation phenomenon impacts signaling. Since the authors suggest that heterotrimers are required for coupling and apparently only Gas translocates it is difficult to see how translocation supports signaling. The second part of the manuscript addresses an entirely different question and looks for active Gas on endosomes using a peptide that binds to Gas-GTP and a nanobody that binds to nucleotide-free Gas. There was virtually no doubt that this happens given prior functional studies and more direct experiments using Nb37 by the same group; the main novelty here is the use of the KB1691 peptide in an endosome assay and the detection of endogenous Gas-GTP after activation of overexpressed receptors. Differences between VIPR1 and A2BR are nice to see but are not particularly surprising given the known properties of these receptors, and Fig. 6 suggests a quantitative rather than qualitative (as implied in Fig. 7b) difference in how they activate G proteins at different locations. One of the most interesting new findings is robust signals from both Gas sensors that persist in Gs/olf DKO cells and are blocked by YM. The authors hesitate to call these signals active G α q/11, presumably because of prior studies claiming specificity of these probes, but "non-Gs" is rather unsatisfying in light of the documented specificity of the YM compound. All in all, the manuscript contains some interesting results, but the large majority of these confirm previous studies, and the conceptual advance is rather limited.

We thank the reviewer for the thoughtful comments. We agree that the first part of the manuscript is not conceptually new, and we don't mean to imply otherwise. Our intention in this part is to place the work in context with present models in the field. We focus here on the β 2 adrenergic receptor because this is the GPCR that has been traditionally used for such studies. We then extend the present framework in 2 ways. First, we show a similar process of Gas redistribution stimulated by two other Gs-coupled GPCRs. Second, we demonstrate that these GPCRs, as well as with the β 2 adrenergic receptor, are able to stimulate a rapid (within minutes) redistribution of Gas from the plasma membrane at endogenous levels of receptor expression. We agree that the Insel group (Ransnas et al, cited) previously provided biochemical evidence for endogenous Gas redistribution stimulated by endogenous β 2AR, but note that this required >1 hour of continuous agonist treatment while we show redistribution within several minutes. Thus it remains unclear whether or how those results would pertain to signaling (or perhaps they were detecting something else) but we simply cite without further discussion because we want to remain focused on redistribution occurring over a signaling-relevant time scale. We believe the present results are the first to show this at endogenous protein levels.

The reviewer raised a concern about the degree of conceptual advance also in the second part of the study, which focuses on G protein activation. We believe that there is considerable advance here and, in fact, that most of what we show in the second part is new. We do agree that the present model holds that Gs activation on endosomes requires a second coupling reaction on the limiting membrane, and that Nb37 recruitment to endosomes supports this model. We claim two key elements of novelty: First, to our knowledge, the accumulation of active-state Gas on endosomes has not previously been demonstrated. We show this in the present study, and at endogenous levels of G protein expression. We think this is significant because G protein coupling and the accumulation of active-state G α subunits on a membrane are mechanistically distinct processes. Second, we demonstrate location bias in G protein coupling on endosomes relative to the plasma membrane. We agree this is perhaps the most novel and interesting aspect of the work, as the reviewer notes, and we have significantly strengthened this part in the revised manuscript.

We detail our responses and changes made in the revision below, in line with specific critiques raised by the reviewer.

Specific comments:

The authors claim that endosomal and plasma membrane signals show expected "sequential" kinetics but there are some anomalies that are difficult to explain. The time courses of plasma membrane and endosome active Gas signals in Fig. 5e and f, and Fig. 6c are not that different; plasma membrane signals rise for several minutes, and endosome signals are more than 50% of maximum at the first time point after VIP addition, even when endocytosis is inhibited. The time course of active receptor appearance in endosomes is much slower. "Non-Gs" signals seem to be much faster and sometimes larger for these Gs-coupled receptors. At the very least a more careful discussion of these issues is warranted.

This is a good point and important to clarify. Yes, we do think that the kinetics are different. The active-state G α s (KB1691) increase at the plasma membrane (Fig 5e) peaks before 10 minutes, but the endosome elevation (Fig 5f) reaches its maximum at about 15 min after agonist application. We agree that active-state G α s accumulation on the endosomes appears to be faster than the endosomal accumulation of active-state VIPR1 (mGs), but there is clearly some active-state receptor in the endosome at the time that we see active-state G protein accumulation. Thus we speculate that this reflects only a fraction of the receptor pool being sufficient to activate the endosomal complement of Gs. We are detecting active-state G α s at endogenous levels, but need to over-express VIPR1 in order to see this, so we think that this is plausible. Nevertheless, there is still a kinetic shift consistent with sequential activation at each membrane location as proposed. We note and briefly discuss this point in the revised manuscript (p. 12 bottom).

We fully agree that the non-Gs component detected at the plasma membrane is indeed faster, and its magnitude is considerable. We find this observation very interesting and note that it would be impossible to detect without the new methodology described in our study. As the reviewer notes, we used the term 'non-Gs' rather than 'Gq/11' in the submitted manuscript. The reason is that we focused primarily on detecting active G α s and, at that time of submission, were not satisfied that we had fully defined the non-Gs component. The simplest interpretation, as the reviewer notes, is that it represents Gq/11 based on its inhibition by YM. We have now verified this conclusion in two ways. First, we have added new data demonstrating a specific loss of the non-Gs component in Gq/11 DKO cells (Supplementary Figs. 8 and 10). Second, we have added new data using an established active-state Gq/11 biosensor (p63RhoGEF, Supplementary Fig. 8), showing that this sensor detects the rapid component of activation consistent with what we previously defined as non-Gs, and this is lost on Gq/11 KO cells. In the revised manuscript we have included these new data as indicated, and have changed to calling this component Gq/11 accordingly.

Dynamin K44E is only partially effective in Fig. 5. Since the receptors here are overexpressed, these experiments should be supplemented with receptors that are completely defective when it comes to endocytosis, and/or additional more effective blockers of receptor endocytosis. In any case the degree to which receptor endocytosis is blocked should be directly quantified.

A related question, why was the β adrenergic receptor ignored for the second half of the manuscript, especially since the 3S mutant is in hand?

We were also surprised that mutant dynamin, while it significantly inhibits both endosomal recruitment of mGs and KB1691, does so only partially for both sensors. The virally expressed K44 mutant dynamin indeed inhibits VIPR1 internalization strongly. We published this previously (Blythe 2024, as cited) and used the same strategy in the present study, but the reviewer is correct that we did not include data verifying this in the initial submission. In the revised manuscript, we have included flow cytometry data to verify this point (Supplementary Fig. 9a). Still, we find only a partial inhibition of mGs and KB1691 recruitment to endosomes as shown. We speculate that this is because of receptor overexpression and that endocytosis concentrates receptors relative to the concentration in the plasma membrane, such that only a small fraction of the overexpressed receptor complement is needed to produce detectable recruitment signal measured by the biosensors. We believe this is the most parsimonious interpretation and are confident that there is significant endocytosis-dependence as claimed. In the revised text (Results section, p.13 bottom of page) we note and discuss this point.

We did not mean to ignore the β adrenergic receptor and, as noted above, we focused on this GPCR in the first part of the paper because it is the GPCR traditionally used for studies of G α s redistribution. We focused on VIPR1 in the second part of the paper for two reasons. First, VIPR1 produces a stronger cAMP response after endocytosis (Blythe et al 2025 as cited). Second, we find VIPR1 more interesting in the context of the G protein activation process because it can stimulate production of both active-state G α q and active-state G α s. We have no evidence for such dual activation by β 2AR and no evidence that the β 2AR exhibits location bias at endosomes.

Regarding the phospho-mutant receptors, in our hands we find that K44 mutant dynamin inhibits β 2AR internalization more strongly than the 3S mutation. We have so far been unable to identify a functional VIPR1 mutant that is endocytosis-deficient using a similar Ser/Thr mutation strategy, and we also cannot use β -arr KO cells because VIPR1 internalization is arrestin-independent. For these reasons, we focused in the present study on the use of K44 mutant dynamin to inhibit receptor internalization.

The authors conclude that VIPR1 activates Gq/11 (non-Gs) on the plasma membrane but not on endosomes, which seems to be the case in Fig. 4f (KB1691) but less convincingly in Fig. 6c (Nb37). If Gq/11 (non-Gs) is present on endosomes (Fig. 6d), how does it avoid activation by VIPR1 at this site?

This is an excellent question, and we presently don't know why VIPR1 fails to detectably couple to Gq/11 on endosomes. We are confident in the biosensor data demonstrating the existence of this additional activation component on the plasma membrane. We speculate that this may have to do with different lipid or protein compositions between the membranes, and this would be potentially in line with differences in G protein membrane binding properties shown years ago by the Wedegaertner lab. However, we have tested this hypothesis in the present and believe that doing so would be beyond the realistic scope of the present study. In the present study, we focus specifically on establishing the existence of this second component of G protein activation and its location-specificity.

The authors conclude in the abstract that activation of endosomal Gas depends on receptor endocytosis, but in fact they only show that this activation is dependent on dynamin. Could dynamin K44E be inhibiting delivery of the peptide ligand to endosome-resident receptors? Experiments with endocytosis-deficient receptors could address this.

We think this is unlikely because mutant dynamin expression does not block pinocytic delivery of fluid-phase markers to endosomes, and there is evidence that it may actually increase flux through this non-clathrin route (PMID 7559787). However, if it were the case that K44 inhibited peptide delivery, it would still be evidence for a role of receptor activation in endosomes. We agree that one could investigate this in more detail with endocytosis-mutant receptors but, as noted above, we have so far not successfully identified such a mutation for VIPR1.

TIRF images show DMSO controls but there are no DMSO traces in the grouped data. Given the small magnitude of receptor-mediated translocation such controls are important in this instance.

This is a good point. We are confident that the observed magnitudes of agonist-induced effects are significant but, in the initial submission, we showed only vehicle-subtracted curves because we thought it made the figure less 'busy'. However, we agree that this is a plausible alternate interpretation in retrospect and, in the revised manuscript, we plot non-subtracted curves and include the vehicle control as the reviewer suggests (Fig. 3 and Supplementary Fig. 7).

The endosome time course traces in Figs. 4 and 5 are missing the first two time points after the addition of agonist, whereas the plasma membrane traces show data for these time points. Why?

This is correct. We systematically excluded the first three time points in the endosome nanobit data, corresponding to the first minute after agonist application. The reason is that we occasionally see a small increase in the KB signal at these very early time points. We believe this cannot be from true accumulation in endosomes because it is seen in the first minute, and that it reflects either a small amount of the FYVE domain (endofin) LgBit construct being mis-targeted to the PM or present on a very early endocytic intermediate compartment. We do not see this consistently, only in some experiments, and thus favor a small amount of mis-targeting. We show representative examples below, with all of the data points included in each case. In the left panel one sees no evidence of a small early elevation, but in the right panel there is a small blip. This observation is 1) sporadic, 2) very low in amplitude and 3) evident when very little receptor is present in endosomes. Therefore we do not think this is biologically significant and, even if it were, our interpretation of the slower (consistent with endocytic dynamics) recruitment to endosomes would not change. In the revised manuscript, we have added a brief explanation of this point in the Methods section (p. 25 middle of page).

Panels 2c and 2d are partially redundant.

The data graphed in each panel were indeed different (corresponding to the same control condition across panels), but we agree they show essentially the same thing. We have removed panel 2C in the revised manuscript for clarity.

Can the authors provide some explanation for why KB1691 was replaced with Nb37 for the later experiments?

We believe that both sensors are important but that each senses something different. KB1691 binds selectively to active-state (GTP-bound) α -subunits (as demonstrated in Janicot et al. 2024, as cited) but Nb37 binds selectively to a nucleotide-free α -subunit conformation corresponding to a catalytic intermediate in the process of GPCR-mediated G protein activation (Westfield et al. 2011 and Irannejad et al. 2013, as cited). Thus we interpret KB1691 recruitment as a measure of active-state α -subunit accumulation on the target membrane and Nb37 recruitment as a measure of the coupling step that mediates G protein activation. The results obtained with

each are consistent, and comport with our present understanding of how G protein activation occurs, but we believe that each biosensor provides useful and distinct information. In the revised text we attempt to clarify this point with more precise language throughout the Results section, and we have added text to explicitly note the difference in our interpretations of KB1691 and Nb37 recruitment results (p. 14 middle of page)

The authors should make a consistent distinction between Gs heterotrimers and Gas subunits. For example, they claim in the abstract that they show that plasma membrane events control endosomal Gs abundance. This should be Gas.

We thank the reviewer for noting these inconsistencies and agree that this is compromised precision. In the revised manuscript, we have corrected this and used consistent nomenclature throughout the revised manuscript.

“Regulate” is a better description than “control” for how plasma membrane activation affects endosome Gas abundance, as the authors do not quantify the extent to which total endosome Gas abundance changes after activation at the plasma membrane.

This is a good point. We have switched to this terminology throughout the revised manuscript.

Reviewer #2 (Remarks to the Author):

Wysomerski et al. Comments

In this manuscript, Wysolmerski and colleagues present data to propose mechanism by which endosomal G(s) G proteins are regulated. As acknowledged by the authors, there is significant accumulated evidence that supports the presence of G proteins in endosomes and G protein activation within these intracellular compartments. Here, the authors use very elegant approaches to show that i) these G proteins “arrive” at the endosomes without independently of the activating receptor but ii) they require an active internalised receptor to be activated. These results are novel and expand our understanding of spatiotemporal GPCR signalling. The therapeutic implications of these findings remain future endeavours, but clearly such knowledge will help in that direction. Of note, during this review, a paper from the same author’s lab has been published and conceptually both papers seem related. It is suggested the authors emphasise novelty of the present work and relationship between both reports.

We are pleased that the reviewer appreciated our experimental approach and found our results generally interesting. The reviewer is correct that we reported data on Gai/o subunit redistribution recently, and we cited this in the initial submission. We think that the properties of Gas are quite different. A key difference is that, in our hands, endosomal accumulation of active-state Gai/o does not depend on endocytosis of the activating GPCR whatsoever. We have emphasized this significant distinction in the revised Discussion section (p.19 middle of page).

I have a few questions about the results presented that require consideration prior to publication. Proposed model of Gs accumulation in endosomes – how are the authors convinced that the Gs accumulated in the endosomes is of PM origin? Is it possible that PM signalling from GPCR/Gs triggers a signal that results in redistribution of Gs from other (non-PM) compartments to accumulate in endosomes? Could the authors perform some tracking of Gs-GFP to assess this?

This is an excellent question and we agree that it is possible, in principle, that the Gas is coming from somewhere else. An ideal approach, as the reviewer suggests, would be to label Gas at the plasma membrane and dynamically track it by imaging. We do this to follow receptors in cells, but we have not successfully achieved this with G α subunits. As an alternative approach to test this hypothesis, we have carried out additional experiments using a mutant Gas that has been shown previously to be N-terminally myristoylated and thereby stably tethered to the plasma membrane (Fig. 1 and Supplementary Fig. 1). Wedegaertner’s group (cited) showed that this modification prevents agonist-induced redistribution of the α subunit to the cytoplasm in fixed preparations. Using live imaging, we verify this and also show that the accumulation of α subunits on endomembranes (including endosomes) is greatly reduced. These data support the hypothesis that endosomal Gas originates from the plasma membrane, but clearly much remains to be learned. These new data are included in Fig. 1c and Supp. Fig. 1a of the revised manuscript, and we have further commented on this important point (adding a new paragraph) in the revised text (p. 6).

The authors suggest that KB1691 recruitment to endosomes, since it is not affected by YM, reflects Gs activation. As they detected a non-Gs component at the PM, one would expect then that this endosomal recruitment of KB1691 is unaffected by Gs/olf knockout?

The reviewer is correct that we detected a non-Gs component of KB1691 recruitment at the plasma membrane, and that we defined this as such because it is unaffected by Gas/olf knockout. The reviewer is also correct that KB1691 recruitment to endosomes is completely lost in Gas/olf knockout cells. This is shown in Fig 4f and we have modified the text in the revised Results section to be more clear on this important point.

KB1691 and Nb37 revealing a non-Gs components – as far as this reviewer understands, both KB1691 and Nb37 were originally described and demonstrated as selective for Gs G proteins. The results here seem to argue against that selectivity and propose that both these sensors can detect a “non-Gs component”. While the data seem to point in that direction, the arguments for “location bias” seem premature without the understanding of what such “non-Gs” mechanism is. With the current justification, this stands out as a major limitation of this work.

We agree that this was a significant limitation in the study as initially submitted. We were also surprised that a non-Gs component was detected (and by both sensors), but were careful to use the term 'non-Gs' rather than 'Gq/11' in the submitted manuscript because, at the time, we only had pharmacological support (YM) for this conclusion. To address this concern we have carried out additional experiments to more fully define the non-Gs component. First, we now demonstrate a specific loss of the non-Gs component in Gq/11 DKO cells, providing genetic evidence for supporting its definition as Gq/11. Second, we have adapted an established active-state Gαq/11 biosensor (derived from p63RhoGEF) and show that this sensor indeed detects the non-Gs component with similar kinetics. Therefore we are now confident the non-Gs component represents Gαq/11, and we have revised the manuscript accordingly.

Other comments:

- While the interpretation of Fig1c is supported by the data, this would be further supported if the authors applied a mask to remove the PM region from their Pearson Correlation Coefficient calculation.

We agree that this would demonstrate a positive change in the Pearson coefficient of accumulation on receptor-containing endosomes as is evident in the individual images. We prefer not to do this, however, because the endosomal colocalization is visually evident in agonist-treated cells and we are concerned that removing the plasma membrane from the region analyzed might introduce unwanted bias to the analysis. We also find it interesting that the Pearson coefficient goes down to an intermediate level after agonist application, and we think that this is consistent with the visual evidence that the redistributed Gαs, while it clearly localizes to receptor-containing endosomes, does not localize there exclusively. Nevertheless, we agree that it would be helpful to provide explicit evidence for an increase in localization at endosomes. To address this, in the revised manuscript we include additional data showing colocalization of β2AR, Gαs and labeled Gβγ on endosomes quantified with line scans (Supplementary Fig. 1c).

- Supp Fig2 and 3 legends have scale bars in microM rather than microm.
- As in ref 33, it would be recommended that they authors refer to the current patent on the KB1691 peptides.
- Fig4 has one panel labelled PM and another Plasma membrane – use one for consistency

We thank the reviewer for noting these errors and have corrected them in the revised manuscript.

Reviewer #3 (Remarks to the Author):

Review of "Conformational biosensors delineate endosomal G protein regulation by GPCRs" by Brian Wysolmerski, Emily E. Blythe, Mark von Zastrow

In their manuscript, Wysolmerski and collaborators shed light on Gs redistribution and activation at the endosome following GPCR activation in HEK293 cells. They find that stimulation of endogenous β2AR, VIPR1 or A2BR causes a loss of Gs from the plasma membrane. In the case of β2AR, they investigated this finding further and found that this loss is independent of receptor internalization and that it corresponds to Gs redistribution to endosomes. In the case of VIP1R and A2BR, they show that endosome-localized Gs is active and, only for VIP1R, they show that this activation is dependent on receptor internalization. Finally, they propose a location bias mechanism in G protein activation by both VIP1R and A2BR, where the proportion of Gs and non-Gs activation is both receptor- and compartment-dependent.

The results are potentially very interesting for the GPCR and cell biology community, however the manuscript suffers from the following limitations:

We are pleased that the reviewer found our results potentially very interesting, and we have done our best to address the concerns noted.

- The conclusions are presented as general mechanisms of GPCR functioning, but these are not supported by the data, which are receptor specific. Moreover, conclusions are based on experiments lacking sufficient number of repeats and from qualitative assessments lacking proper quantification.

We agree that we cannot claim a general mechanism for all GPCRs, and that this is not supported by our data. In fact, our data show that Gαs redistribution appears rather general but that there are differences among receptors in their coupling at different membranes. We also acknowledge that some of the supplemental data were only n = 2. We have corrected this in the revised manuscript and added quantification in all, as suggested.

The approaches used are not so novel. Nb37, NanoBit assay, KB1691 have been already described and used in several occasions (mentioned in the text by the authors). However, Nb37 and KB1691 here are used in a clever way to distinguish between GPCR-bound G protein versus active (=GTP bound) G protein. Similarly, the NanoBit assay was adapted to these specific questions.

We are pleased that the reviewer found our adaptation of the biosensor approach interesting, and agree that the basic elements were previously established (we specifically state this and include relevant citations). The important new aspects are 1) location-specific detection, 2) detection of G protein activation and active-state Gα accumulation at endogenous expression levels, and 3) detection of location-biased coupling. To our knowledge none of this has been established previously.

- I don't see the relevance of the finding that G proteins are redistributed to endosomes independently of receptor endocytosis. What does it mean physiologically speaking? What is the function of Gs redistribution to the endosome in absence of receptor?

We agree that the ability of Gas to redistribute without endocytosis of the activating receptor is interesting, but also note that this has been described previously. Our results verify and support this conclusion using new methods, relative to earlier reports. We think it important to establish this in our system because the production of active-state Gas on endosomes differs in being endocytosis-dependent (Fig. 5). The functional significance of Gas redistributing separately from activation is an excellent question that our study does not address. Our study focuses on establishing the difference that the reviewer notes, and revealing the additional and unanticipated selectivity in active-state G α production. One can speculate that Gas redistribution without receptors might have indirect effects by changing availability to other GPCRs, but we presently have no data to support or refute this. In the revised manuscript, we have tried to clarify the focus of the present work and note that the functional significance of the distinction that the reviewer notes remains to be determined (p.18).

- The finding that G proteins are active at the endosome only when the receptor is also there is somehow reassuring, but expected I would say.

We agree, although note that there is evidence for an additional mechanism of activation that does not appear to work as expected, for Gi/o-class G proteins (Fisher et al. 2025, as cited), so we think it is worthwhile establishing that Gs indeed appears to work as expected.

- No downstream effect of location bias in G protein activation by VIPR1 and A2BR has been assessed.

This an excellent question that we have not yet addressed. Our goal in the present work is to establish the location of G protein activation by these GPCRs and we believe the new data included in the revised manuscript help establish the existence of this location bias. However, we do not yet know the effect(s) of this bias on downstream signaling. We have explicitly noted this point in the revised Discussion section (p.19 top of page).

- No information on how G proteins and receptors couple at the endosome is provided. Do they traffic together? Is the receptor internalized first and then the G protein recruited from the cytoplasm? Is through fusion of 2 endosomal compartments (one containing the receptor and one showing G protein presence)?

We do not think that Gas traffics together with the activated receptor. We show evidence supporting this conclusion (Fig. 2 and Supplementary Fig. 3) and also cite previous work from others consistent with it. We believe that the membrane trafficking model suggested by the reviewer is plausible, but another plausible model is that Gas partitions in and out of membranes to dynamically sample intracellular compartments after activation (this is a model supported also by Martin and Lambert 2016 as cited). Our results do not distinguish these possibilities. Rather, our study focuses on 1) demonstrating that multiple Gs-coupled GPCRs are capable of stimulating dynamic redistribution of Gas, 2) that they can do so at endogenous levels of receptor and G protein expression, and then 3) on probing the subcellular locations of Gs activation and membrane accumulation of active-state Gas. In the revised manuscript we have attempted to better clarify this focus, and note in the discussion that there remain significant unresolved aspects of Gas trafficking (p. 18).

- Only G protein redistribution has been shown to happen at native levels of receptor, not G protein activation. The authors should reconcile these findings.

This is correct. Our biosensor strategy is capable of detecting G protein activation and active-state α -subunit accumulation at native levels of G protein expression, but we have so far been unable to detect this after activation of an endogenously expressed GPCR. This is the case even for activation at the plasma membrane, so we believe that it reflects a limitation in sensitivity of our probes. A main new advance in the present study is detecting G protein activation in a spatially resolved manner (at endogenous G proteins but admittedly overexpressed GPCR), although we agree that it is important to continue working toward achieving such detection in a fully endogenous system. In the revised manuscript, we attempt to reconcile and better clarify this issue by emphasizing that the present need for receptor overexpression, despite clear evidence (in previous work from others and us) that receptor signaling from multiple membrane locations indeed does occur (starting on p. 11 middle of page).

Apart from these concerns, please see below for other comments that should be addressed before the manuscript is suitable for publication:

1. p.3 "However, the presence of active-state Gas on endosomes has only been hypothesized." I guess the authors mean GTP-bound G protein. I this needs to be clearly stated as it can be perceived that all the work done by von Zastrow's lab and others using MiniG proteins and Nb37 does not point in that direction.

We agree that the evidence strongly points in this direction, and that our wording in retrospect was a bit strong. Indeed we and others have detected membrane recruitment of Nb37 previously. We presently understand this probe to detect an intermediate in the process of Gas activation (binds selectively to nucleotide-free state), so we consider Nb37 to represent a sensor of coupling rather than active-state Gas (Nb37 binds poorly to the GTP-bound Gas). We think KB1691 is useful because this probe has been shown to specifically detect the true active-state (GTP-bound) Gas. In the revised manuscript, we have clarified this distinction between the probes (p.14 middle of page).

2. p.3 “significantly differ in their ability to internalize after activation.” Is this a finding from this study or something already known? If known, please add refs, if new... I don't see a characterization of the 3 receptors internalization in this paper, unless I missed it. We thank the reviewer for noting this. β 2AR is well known to internalize via clathrin-coated pits and in an arrestin-dependent manner. We recently reported that VIPR1 internalizes even more rapidly, also via clathrin-coated pits, but in an arrestin-independent manner (Blythe et al 2024 as cited). The lack of detectable internalization of $A_{2B}R$ is demonstrated in another manuscript that is presently in revision, and we cited it as the Blythe et al BioRxiv posting. We do not claim an absolute lack of receptor endocytosis and, in the revised manuscript, we include an additional image (Supplementary Fig. 4b) showing that human $A_{2B}R$ remains largely in the plasma membrane after agonist application, and we have more explicitly indicated the basis of our descriptions in the revised text (p. 9 middle of page).

3. In Fig. 1, colocalization between G and EEA1 is missing. Fig.1d is only representative/qualitative but no quantification is shown for Gas G colocalization at the EE.

The reviewer is correct, we label no more than three proteins in any image. We would like to label all four proteins noted simultaneously, but are presently unable to image more than three channels without having to correct for artifacts of fluorophore spectral overlap (which we assiduously avoid for examination of colocalization). In considering the reviewer's comment, we recognize that we failed to explicitly show $G\beta\gamma$ colocalized with internalized receptors and Gas. We can assure the reviewer that this is the case and apologize for missing this point. In the revised manuscript we included (Supplemental Figure 1d) an image of the internalized receptor, Gas, $G\beta\gamma$. We also show a representative line scan of each channel. One can clearly see that Gas and $\beta\gamma$ overlap the internalized receptor channel (and these are truly internalized receptors based on the labeling method used); however, Gas and $G\beta\gamma$ are not restricted to receptor-containing endosomes and, in fact, are quite broadly distributed. This is consistent with previous studies from others describing endogenously labeled G protein localization (from the Lambert group) and its subcellular distribution by cell fractionation (from the Leonetti group; including specifically with EEA1). In the revised manuscript we note this broader distribution and cite these additional studies that support our conclusion (bottom of p. 6 and top of p. 7).

4. Also in Fig. 1 I'm surprised not to see G at the PM in unstimulated cells. I was expecting complete colocalization with Gs. Could the authors comment on this?

This is an excellent point. Actually we do see evidence for $G\beta\gamma$ localization to the plasma membrane, as the reviewer predicts. However, it is not only at that location and it is generally easier to visualize punctate internal structures using confocal microscopy. PM localization is evident by line scan analysis, however (an example is shown in the new data in Supplementary Figure 1d). We also note that $G\beta\gamma$ has previously been shown to localize intracellularly in the absence of receptor activation (Cho et al. 2022, Jang et al. 2024, as cited), and that differences in the relative localization of distinct $G\beta\gamma$ combinations have been observed (Masuho et al. 2021, as cited). While our results do not go into this level of detail, we believe our results are fully consistent with previous studies and have emphasized this in the revised text (bottom of p. 6 and top of p. 7).

5. Could the experiment performed in Fig. 2 be done in a complementary way using LgBit-Gs and SmBit-endofin, to look at redistribution of Gs to and from the endosome upon Iso or Alp addition?

Yes we think so. We focused on assessing loss from the plasma membrane in Fig 2 because this is the result that we sought to verify, in context of the prevailing current view that Gas dissociates from the plasma membrane and its increased concentration in the cytoplasm drives 'sampling' of intracellular membranes. However, the microscopy data indicate that this sampling process is not endosome-specific, as other internal membranes also visibly recruit Gas. A concern with the complementary approach is how one would interpret an increase in nanobit signal. An increase of the nanobit signal on endosomes could be explained simply as a consequence of this increased cytoplasmic concentration. This is not an issue in the endosome nanobit assays using G protein activation sensors because their concentration in the cytoplasm does not increase with receptor activation. If anything it decreases, due to membrane recruitment, and thus an increase in the endosome signal is not subject to the same caveat.

6. In Fig. 2e, redistribution of Gs from the endosome to the membrane upon Alp addition is observed, even in absence of receptor internalization. It would be great to hear what the authors think about this finding, if they could speculate on how Gs can sense from the endosome the presence of an antagonist acting on receptors blocked at the plasma membrane.

This is an interesting idea. The reviewer is correct that Alp is membrane-permeant and we have not ruled out communication back from the endosome, as the reviewer proposes. However, we don't presently think that retrograde communication would be needed to

explain the observed return of Gas to the plasma membrane after Alp addition. The reason is that we believe Gas dissociates from the plasma membrane after receptor activation and reassociates through a cytoplasmic intermediate form (this goes back to the early observations of Rasnas et al, and Rodbell even before that, both cited). If recovery of Gas to the PM required retrograde communication, we would not expect the result shown. Instead, we think that antagonist application simply reverses the dissociation process from the PM, and this does not require any communication from the endosome.

7. Why did the authors choose to block endocytosis of VIPR1 by overexpressing the dominant negative mutant of dynamin Dyn1-K44E instead of using Dyngo-4a as it was done for β 2AR?

Both Dyngo-4a and Dyn-K44E are useful inhibitors of endocytosis, and we generally observe similar results with both. We used Dyn-K44E in the studies noted because we found Dyngo-4a to variably suppress the luminescence signal in our NanoBit assays. We believe this to be a technical off-target effect of Dyngo-4a, somehow it seems to variably inhibit some aspect of the luciferase reaction. We have found Dyn-K44E to be preferable for NanoBit experiments because its expression does not affect the luminescence signal.

8. Please add quantification for confocal images shown in Supp. Fig. 1, 2, 3 and 4. The analysis should be conducted on a bigger sample, not just 4 cells, from more than 2 independent experiments.

We thank the reviewer for noting this lack of quantification and have added line scans to quantify the images in Supplementary Figures 1 through 4. We have also added a third replicate of the experiments in Supplementary Figure 2b and Supplementary Figure 4. For experiments in Supplementary Figure 4b (A2bR images), we repeated the imaging in the presence of a PKA inhibitor (H89) to prevent contractility induced by activation of overexpressed $A_{2b}R$ and improve the quality of images. The adenosine receptor images have been replaced to reflect this experimental change, and we note this in the methods and figure legends (Methods, p. 22 and Supplementary Fig. 4).

9. More repeats should be included for experiments that were repeated only twice. 4 cells are not sufficient. I suggest analyzing 5 cells per repeat, for a total of 15 cells.

We agree with the reviewer and thank them for pointing this out. The TIRF experiments in Figure 4 and Supplementary Figure 7 represent data from at least 4 individual movies, with 2-6 cells per movie. F/F_0 values for each cell within an individual movie were averaged to calculate average F/F_0 values for each individual replicate, so for all conditions, we analyzed at between 10 and 20 cells. We had forgotten to include this information in the figure legends and methods and have added it to the revised manuscript (p. 24, middle of page, and Fig. 4 and Supplementary Fig. 7 legends).

10. There is a clear imbalance between the first part of the manuscript where all experiments are conducted using β 2AR as model GPCR and mainly live-imaging techniques, and the second part of the paper where experiments are focused on VIP1R and A2BR using the NanoBit assay.

Specifically, the authors show that Gs redistribution to the endosomes is independent of receptor endocytosis only for β 2AR. It would be informative to know whether this is also true for other Gs coupled GPCRs such as VIPR1 and A2BR.

Similarly, was the NanoBit assay run for β 2AR with either KB1691 or Nb37? The manuscript would benefit from adding these data.

These are all fair points and, in considering them, we recognize a need to further clarify our experimental focus and rationale.

Our claim in the first part of the paper is that Gas redistribution from the plasma membrane does not require internalization of the activating GPCR. Our rationale for emphasizing the β 2AR was 2-fold: First, this is the GPCR traditionally used for studies of Gas redistribution, and our intention was to place the present results in context with the current understanding developed largely through the study of β 2AR-induced effects. Second, the β 2AR is experimentally advantageous because we have three independent methods for blocking its internalization (β arr DKO, β 2AR-3S mutant and dynamin inhibition). We believe that $A_{2b}R$ provides further support for our conclusion because the human $A_{2b}R$ naturally does not internalize after agonist-induced activation; thus it enables us to assess Gas redistribution in the absence of any experimental perturbation of trafficking. We would prefer not to make a strong claim for VIPR1 because we have only one manipulation (dynamin inhibition) to block its internalization (it is not blocked by β arr DKO (Blythe et al. 2024) or phospho-site mutations). In addition we think that, even if the properties of Gas redistribution triggered by VIPR1 were somehow different, such a difference would not change our conclusion that internalization of the activating GPCR is not required.

Our claim in the second part of the paper is that Gs activation on the endosome membrane (assayed by the Nb37 biosensor) is dependent on the presence of activated receptors in the endosome, and that it produces an accumulation of active-state G α s on the endosome membrane (assayed by the KB1691 biosensor). We previously reported that β 2AR activates Gs on endosomes using a microscopy-based Nb37 recruitment assay (Irannejad et al. 2013). We focused here on VIPR1 because this GPCR more robustly stimulates Gs signaling from both the plasma membrane and endosomes (Blythe et al. 2024). The β 2AR can also activate Gs at both locations, but it does so less strongly than VIPR1 (Blythe et al. 2025). Therefore we have focused on VIPR1 in the present study, and believe that our results support the claimed conclusions.

In the revised manuscript, we have clarified our focus on the cell biology of Gs, rather than differences among the various Gs-coupled GPCRs we used to interrogate it. We explicitly acknowledge that there may be additional GPCR-specific biology that is not uncovered by our present work. We further clarify that our only strong claim about GPCR-specific differences exclusively concerns the last part of the manuscript, where unexpected differences in G protein coupling selectivity at the endosome membrane relative to plasma membrane are described.

Finally, since KB1691 is conveniently tagged with mApple, it would be good to include confocal images of KB1691 localization before and after stimulation with either VIP or NECA (+ localization markers for PM and endosomes).

This is a good point. We were unable to detect KB1691 recruitment visually by endogenous G proteins using confocal microscopy, at either the plasma membrane or endosomes. We think this is a detection sensitivity issue because we know that G protein activation occurs at both locations (previous studies with Nb37). This was the motivation for developing the NanoBit assay, which we find is more sensitive.

11. In Fig. 7a, the authors talk about a general mechanism, but they demonstrate that redistribution of Gs to endomembranes is independent of receptor endocytosis only for β 2AR. Furthermore, the representative GPCR is in purple, which in 7b represents VIPR1, for which receptor independent Gs redistribution to endosomes has not been shown. The scheme is confusing from different aspects and I recommend changing it to reflect the findings described in the manuscript.

We thank the reviewer for this feedback. Our goal in Fig 7a was to provide a general framework, but we agree that we only support this explicitly with 2 of the three GPCRs used in the present study (β 2AR based on three independent endocytic inhibitors, and A2BR based on its natural resistance to agonist-induced internalization). We agree that our initial scheme was confusing and may have overstated the generality of the framework depicted. We do not wish to imply that all GPCRs behave the same way (and the G protein coupling selectivity data indicate that they do not). In the revised manuscript, we have changed the figure to more closely reflect our findings and changed the color of the GPCR in Figure 7a to avoid the potential confusion noted by the reviewer. We have also clarified in the text (p. 18) that there may exist still more GPCR-specific differences that are not uncovered by our present data .

Minor comments:

12. Since in some experiments proteins are overexpressed and in some other experiments not, it would be cleared to add on certain figure panel (e.g. Fig 1a) which tag was used, so it's clear when proteins where overexpressed (e.g. Flag-b2AR, Gs-EGFP, etc).

We agree with the reviewer that this would improve clarity and have revised the figures to include the tag name to clarify when a protein was overexpressed.

13. It is sometimes difficult to understand when the authors mean G protein activation or G protein-GPCR coupling. From what I read, G protein activation is monitored with KB1691, while G protein-GPCR coupling with Nb37. If that's correct, I suggest changing Supp. Figure 9 graph headers to (G protein-GPCR coupling, Nb37).

This is correct: we interpret KB1691 as a probe for the active-state (GTP-bound) $G\alpha$ subunit, as established previously by Janicot et al. as cited. We interpret Nb37 as a probe for the G protein coupling reaction because Nb37 binds selectively to a nucleotide-free α -subunit conformation corresponding to the catalytic intermediate stabilized in GPCR-G protein coupling complex, as established previously by Westfield et al. as cited. We have clarified this distinction in the revised manuscript and figures, and we have attempted to correct any additional inconsistencies in our language when describing the biosensor results (see Supplementary Fig. 10 and p. 14 middle of page).

14. Supp fig. 2a please add single channel images in black and white.

We have added these images to the revised manuscript (Supplementary Figure 2a).

15. The quality of some images is quite poor. Would it be possible to replace images in Fig. 1d +iso 20' and Supp Fig. 4b with better ones?

We have added a second example of $G\alpha_s$ + $G\beta\gamma$ images (Supplementary Fig. 1d) and replaced the images in Supplementary 4b. As noted above (point 8), we found during the revision process that PKA inhibition (by pretreatment with H89) vastly improved the quality of imaging with overexpressed $A_{2B}R$ as, in the absence of PKA inhibition, activation of overexpressed $A_{2B}R$ leads to a large contractile response. We have updated the revised text to reflect this change (see Methods p. 22 and supplementary Fig. 4 figure legend).

16. p.9 "Signaling from endosomes requires $G\alpha_s$ associated with the endosome membrane to be in an active state." Please add a reference to this sentence.

We thank the reviewer for noting this omission. We have clarified this statement as "The current understanding holds that Gs-mediated signaling requires $G\alpha_s$ to be in its active (GTP-bound) conformation" and have added two references supporting this (p. 10).

17. Personally, I do not like the word "abundance" used throughout the text, as to me, it implicates some quantitative meaning, while here it means purely presence of Gs. If the authors agree with my logic, could this word be changed to "presence" or "redistribution"?

This is a great point and we have changed the terminology throughout the revised manuscript and no longer use the word "abundance".

Reviewer #4 (Remarks to the Author):

In this study by Wysolmerski et al., the authors reexamine previous findings related to Gs trafficking and endosomal activity and provide the first results exploring endosomal signaling with the new Gas-GTP sensor from the Garcia-Marcos lab. These results are timely because researchers studying location bias and intracellular signaling have surely thought about adapting this biosensor to fit their research questions since its publication last year. It is certainly exciting to see that this biosensor, when adapted to measure endosomal signaling responses, can detect robust signals at endogenous G protein levels. However, the authors present some curious findings related to the specificity of two biosensors at the center of their study (KB1691 and Nb37). It feels as though they are concluding that the detection modules are not selective for Gas while not ruling out the possibility that other types of G protein activation may affect the concentration of components on endosomal membranes. This distinction is important because the authors conclude that there is a non-Gs component being activated at endosomes for receptors like A2BR when it may be that Gs is indeed activated at endosomes, but Gq/11 activity at the plasma membrane modulates the possibility for Gs to be activated subsequently at endosomes.

We thank the reviewer for these thoughtful comments, and we are pleased that this reviewer found our results interesting and timely. We were also surprised to detect a non-Gs component with both KB1691 and Nb37. We agree that the interpretation of this component is important and have made efforts to improve this, as detailed below.

My suggestions can be found below:

1. YM acts as a molecular glue keeping Gαq and Gβγ together as an inactive heterotrimer (Mühle et al., PNAS, 2025). Can the authors rule out that YM treatment doesn't sequester Gβγ from Gas? Have the authors performed the same experiments in Gαq/11KO cells? In a similar study from Vilardaga and colleagues (White et al., PNAS, 2020), Gαq/11 knockout cells were used and the authors observed a decrease in endosomal cAMP generation which they proposed was dependent on the liberation of Gβγ and PIP3 production to promote GPCR-arrestin binding (in this instance PTHR) and internalization. In the Cell paper from Garcia-Marcos and colleagues where KB1691 was described for the first time (Janicot et al., Cell, 2024), I do not think any Gq/11 inhibitors or knockout lines were used in the context of this sensor; their experiments to confirm G protein selectivity were based on representative tagged G proteins from the four major families and only tested PTX and YM on Gi/o and Gq/11 respectively. I think that the authors from the current study need to provide more convincing evidence that this sensor is detecting something it is not, rather than the consequence of events that preceded downstream signaling at endosomes. For example, it could still be that KB1691 and Nb37 are Gas-selective, but that endosomal Gas activation is affected by Gq/11 activity at the plasma membrane – something that may be receptor-specific.

We agree with the reviewer that there are a number of possibilities for the origin of the non-Gs component. We do not think that it represents indirect activation of Gas (such as by Gαq or βγ) because the non-Gs component is unaffected by Gas/olf knockout (Fig. 4 (KB1691) and Supplementary Fig. 10 (Nb37)). As the reviewer notes, we provided pharmacological (YM) but not genetic evidence for defining this component as Gαq/11, and thus stuck with the more conservative term. In the revised manuscript we include the following new data to address this: 1) We show that the non-Gs component is lost in Gαq/11 knockout cells (Supplementary Figs. 8 (KB1691) and 10 (Nb37)); 2) We further support the pharmacological definition by showing that YM has no effect on sensor recruitment in the Gαq/11 knockout cells (Supplementary Figs. 8 and 10); and 3) We adapt an established active-state Gαq/11 binder to provide further support for the conclusion that the non-Gs component of biosensor recruitment indeed represents the production of active-state Gαq/11 (Supplementary Fig. 8). By including these elements of new data, we now feel comfortable defining the non-Gs component as Gq/11. We have modified the Results section to incorporate these new data and now explicitly conclude Gq/11 as the reviewer suggests.

2. While the results with VIPR and DynK44E suggest that endosomal production of active state Gas is endocytosis-dependent, there could be other factors that regulate the presence of Gas-GTP at endosomes like phosphodiesterases, RGS proteins and intrinsic GTPase activity. In other words, if VIPR internalization could be blocked completely, would we still observe residual endosomal signaling and what could that be due to? Consider other receptors that have a greater dependence on arrestin/CME like V2R. Does this sensor have the dynamic range to measure the shaping of endosomal responses by other proteins like PDEs and intracellular RGS proteins?

We agree that there are likely other factors influencing G protein activity on endosomes. We also found it remarkable, as the reviewer points out, that DynK44E does not fully block the detection of active-state VIPR1 or of active-state Gas at endosomes. We do believe that fully preventing active-state VIPR1 production in endosomes would block the accumulation of active-state Gas on the endosome membrane. However, we have not been able to explicitly test this hypothesis because, while DynK44E strongly inhibits agonist-induced internalization of VIPR1 (as shown previously in Blythe 2024 as cited and now verified in the present study by flow cytometry in Supplementary Fig. 9), there is clearly still some receptor activity in endosomes under this condition. We also explicitly note this in the revised Results section (p. 13 bottom of page).

3. Fig. 1 Gγ2 is geranylgeranylated and does not leave the membrane as readily as other gamma subunits (Masuho et al., Cell Systems, 2021). While there is a clear intracellular localization of Gγ2, the conclusion that this population of Gβγ supports endosomal signaling is probably isoform-specific. In fact, it has also been proposed that Gβγ dimers internalized with β-arrestin-bound receptors can enable a second round of Gas activation for select receptors (Sokrat et al., Communications Biology, 2024). Given these previous findings, I suggest that the authors either broaden their conclusion about how agonist-induced activation happens at endosomes or perform additional G gamma experiments with other subtypes or receptors.

This is an excellent point and we recognize, in retrospect, that we overstepped in our conclusion and have revised it accordingly, stating that only that G β γ is associated with endosomes, and is evident both in untreated and agonist-treated cells, but make no conclusion about whether this association is agonist-regulated (bottom of p.6 to top of p.7). We also thank the reviewer for noting Sokrat et al and have modified the Discussion section to include this citation (p.18, middle of page) and to note that much remains to be learned about G protein trafficking.

4. Fig. 1C label EGFP as control or cytosolic EGFP. Same for Fig.5 and mCherry.
We have corrected this.

5. It is unclear how subpanel 2C differs from the wildtype response in panel 2D. I would suggest reorganizing the subpanels to avoid duplication of results within the same figure because it is essentially the same experiment for WT in subpanels 2C and 2D. These are not the same data but we agree they show the same basic result. In the revised manuscript, we have removed panel 2c for clarity.

Response to Reviewers

We are pleased that reviewer 1 recognizes that we can now assign the YM-sensitive component definitively. This required including additional data using Gq/11 KO cells and also developing an orthogonal biosensor strategy for detecting active Gq/11 at endogenous levels. Reviewer 1 is correct that a number of results shown are generally consistent with previous work (including our own). We indicate and cite this explicitly, and include it only to place our new results in context with previous work. This reviewer appears not to recognize other new contributions of our study: 1) that the previously described Gas redistribution process precedes coupling on endosomes and is triggered by a discrete GPCR - G protein reaction at the plasma membrane; 2) explicit evidence that coupling on endosomes requires endocytosis of the receptor (the reviewer appears to think this is obvious but it is explicitly not the case for Gi as shown in PMID 40261932); 3) identifying a type of location-bias switching in the selectivity of G protein coupling on endosomes relative to the plasma membrane; and 4) demonstrating that this location-bias of endosomal G protein activation is itself GPCR-specific.